# S-Palmitoylation of Synaptic Proteins in Neuronal Plasticity in Normal and Pathological Brains

**DOI:** 10.3390/cells12030387

**Published:** 2023-01-21

**Authors:** Anna Buszka, Agata Pytyś, Domnic Colvin, Jakub Włodarczyk, Tomasz Wójtowicz

**Affiliations:** Laboratory of Cell Biophysics, Nencki Institute of Experimental Biology, Polish Academy of Sciences, 02-093 Warsaw, Poland

**Keywords:** S-palmitoylation, synaptic plasticity, learning and memory, brain disorders

## Abstract

Protein lipidation is a common post-translational modification of proteins that plays an important role in human physiology and pathology. One form of protein lipidation, S-palmitoylation, involves the addition of a 16-carbon fatty acid (palmitate) onto proteins. This reversible modification may affect the regulation of protein trafficking and stability in membranes. From multiple recent experimental studies, a picture emerges whereby protein S-palmitoylation is a ubiquitous yet discrete molecular switch enabling the expansion of protein functions and subcellular localization in minutes to hours. Neural tissue is particularly rich in proteins that are regulated by S-palmitoylation. A surge of novel methods of detection of protein lipidation at high resolution allowed us to get better insights into the roles of protein palmitoylation in brain physiology and pathophysiology. In this review, we specifically discuss experimental work devoted to understanding the impact of protein palmitoylation on functional changes in the excitatory and inhibitory synapses associated with neuronal activity and neuronal plasticity. The accumulated evidence also implies a crucial role of S-palmitoylation in learning and memory, and brain disorders associated with impaired cognitive functions.

## 1. Introduction

The brain has a remarkable ability to adapt to external and internal stimuli. At the cellular level, brain cells undergo functional and structural changes which require synthesis of new proteins. Post-translational modifications (PTMs) may occur after or during the process of protein synthesis [1]. There are more than 300 types of PTMs in the cell, and new ones are still being discovered [2]. PTMs are mainly of two types: proteolytic or autocatalytic cleavage of the peptide from protein [3]; covalent additions of some chemical group to the amino acids, including: protein acetylation, phosphorylation, and methylation. Moreover, the addition could also involve a polypeptide chain, as in ubiquitination or SUMOylation; or other molecules, as in acylation, oxidation and other PTMs [4,5]. It has been hypothesized that PTMs acquired during animal evolution give proteins new features which are necessary for the evolution of complexity and capabilities of the nervous system [6]. Consequently, PTMs of synaptic proteins are found ubiquitously, yet they represent a discrete molecular switch enabling the expansion of protein functions and localization. PTMs may support synaptic function, synapse plasticity and ultimately higher brain functions. Several excellent reviews have described the roles of various PTMs in brain functions [7,8,9]. In this review, we discuss recent experimental work devoted to understanding the impact of protein S-palmitoylation on the function of the excitatory and inhibitory synapses and synaptic plasticity in the central nervous system. We also discuss to what extent S-palmitoylation of synaptic proteins may be linked to learning and memory and the pathomechanism of several brain disorders and diseases in which aberrant neuronal plasticity and impaired cognitive functions are observed.

### 1.1. Lipid Post-Translational Modifications

Lipid modifications are a category of PTMs which involve the addition of a lipid or a lipid-like group onto proteins. This process increases the hydrophobicity of the modified protein, allowing it to integrate with the lipid bilayer in the cell, which makes it a vital element of protein trafficking and stability in membranes [10]. Lipid modifications represent the group of fatty acid acylation processes. There are several types of lipid modifications based on the length of the carbon chain in the lipid molecule and the type of amino acid in the structure of the protein to which the lipid molecule is attached (Table 1) [10,11]. S-acylation is a type of lipid modification in which fatty acid (chain length ranges between 14 and 20 carbons) is added to cysteine residue of the protein by thioester linkage. Fatty acids of varying lengths of the chain could be added in this process: myristate (14:0), palmitate (16:0), stearate (18:0), and oleate (18:1) [9]. In a primary culture of rat cortical neurons, palmitic, stearic, and myristic acid may constitute 86% of all fatty acids [12]. Similarly, in non-neuronal cells (mouse muscle cell line BC3HI and human epidermoid carcinoma A431), palmitate was reported to account for approximately 75% of the protein-bound fatty acids (from which circa 50% was designated as an ester-linked form), and myristate and stearate contributed to the remaining 25% [13]. Other studies in platelets and the COS-1 cell line also implicated palmitate as the predominant fatty acid bound to peptides [14,15]. As palmitate is the most common lipid group added to cysteines in the process of S-acylation, the process is also commonly referred to as S-palmitoylation, or simply palmitoylation.

### 1.2. DHHC Family of Palmitoylation Enzymes

#### 1.2.1. Enzymes Nomenclature and Structure

Since the discovery of the S-acylation process in the vesicular stomatitis virus in 1979 [16], it took over 20 years to find enzymes responsible for this process. Research in *Saccharomyces cerevisiae* showed that palmitoyltransferases (PATs) contained a conserved sequence motif Asp-His-His-Cys (DHHC) in the cysteine-rich domain (DHHC-CRD), which was shown to be essential for the activity of the enzyme [17,18]. It led to the search for other PATs belonging to the DHHC-CRD family, as it was believed that they were genetically conserved PATs [19]. According to the newest nomenclature, PATs are referred to as zinc finger DHHCs (ZDHHCs) [20]. In humans, there are 23 of them described with numbers from 1 to 24 (skipping the number 10), and they belong to the family of membrane proteins. Apart from the highly conserved DHHC motif in their structure, all ZDHHC enzymes contain at least four transmembrane domains (TMDs), and in most cases, the N- and C-termini are located on the cytoplasmic side of the membrane. In contrast to the conserved motif, the terminal domains vary a lot in length and sequence [21]. The enzymes utilize a two-step “ping-pong” mechanism. The first step involves auto-acylation with the fatty acyl group. This fatty acyl group is then transferred to a cysteine residue of ZDHHC, creating an intermediate state of palmitoyl-ZDHHC which can relocate the palmitoyl group to a cysteine residue of a substrate protein. Individual ZDHHC enzymes have marked differences in fatty acid selectivity, and the underlying molecular basis for this property has been described [22]. Using mutagenesis, the authors discovered that one amino acid (I182) present in the transmembrane domain is responsible for the acyl-CoA preference of ZDHHC3. This was further evidenced for zebrafish DHHC15 and human ZDHHC20 [23]. The ZDHHC structure and the catalytic mechanism are described in detail in [24].

#### 1.2.2. Localization

A single study describing precise localization of all ZDHHC in the brain cells is lacking. In a seminal study by Levy et al., neuronal localization of nine different palmitoyl transferases was shown to predominate either in Golgi or neurites [25]. Below, we mention studies reporting neuronal localizations of ZDHHCs with particular emphasis on synaptic and dendritic compartments. PATs may be localized to the plasma membrane thanks to the signaling motifs in their C-terminal tails that control their recruitment to the cell membrane [26,27]. For instance, in primary cultures of hippocampal neurons, ZDHHC2 was localized in dendrites near the postsynaptic membranes [27] and was also shown to co-localize with a recycling endosome marker in dendrites [28]. Localization in dendritic vesicles and dendritic spines was also observed for ZDHHC5 in primary hippocampal neurons [29]. At least two palmitoylating enzymes, ZDHHC2 and ZDHHC5, were shown to operate within dendritic spines and dynamically palmitoylate substrates upon altered neuronal activity [26,28,30,31]. The role of ZDHHC5 in activity-dependent synaptic plasticity has been recently reviewed by [32]. ZDHHC8 was also shown in cultured hippocampal neurons to be present in vesicle-like clusters along the dendritic shafts [33]. ZDHHC23 (DHHC11 in the old nomenclature) was found in the synaptic plasma membrane fraction obtained from a rat brain and in synaptic sites in primary cultures of cortical neurons [34]. Brigidi et al. also reported that in hippocampal neuronal cultures, ZDHHC5 is present in 80% of excitatory and 47% of inhibitory synapses [26]. Similarly, ZDHHC9 and ZDHHC15 were discovered in particles apposing excitatory and inhibitory synaptic markers [35,36]. In primary hippocampal neurons, ZDHHC12 was shown to have Golgi localization in the cell body and dendrites as vesicle-like structures [37]. While most ZDHHCs have been ascribed to somato-dendritic compartments, relatively little is known about the presynaptic location. *Drosophila melanogaster* PATs Huntingtin-interacting protein 14 (HIP14), which is a homolog of human ZDHHC17, was localized to the pre- and postsynaptic regions at the neuromuscular junction [38]. Moreover, this human homolog was also shown to localize to presynaptic terminals in flies and cultured rat hippocampal neurons [39]. Altogether, the localization of ZDHHCs in somato-dendritic and synaptic compartments suits them ideally for compartment-specific modification of neuronal proteins. A picture emerges that individual members of palmitoylation machinery may be in close vicinity to synaptic sites to support synapse function and plasticity.

#### 1.2.3. Substrates

The domains located in the C-terminal part of the ZDHHCs are believed to be involved in the substrate specificity of the enzymes and shown to play vital roles in substrate recruitment [40]. Some proteins could be palmitoylated by more than just one ZDHHC, indicating redundancy. However, there are also examples of unique selectivity of enzymes. For example, it was shown that only one of the three isoforms of the plasticity-related gene 1 (PRG-1) protein is palmitoylated in vitro [41]. The same phenomenon was also shown for other proteins [42,43]. Moreover, ZDHHC2 and ZDHHC3 were shown to have lipid-substrate-specificity [44]. Fukata and et al. described an approach which allows one to establish which PATs are responsible for the modification of a given substrate protein [45]. This method was also successfully used to discover enzymes palmitoylating postsynaptic density protein 95 (PSD-95), which plays a crucial role in the organization of ionotropic receptors and many other proteins at the postsynaptic site of excitatory synapses [43,46]. Likewise, many neuronal proteins, including synaptic proteins, are palmitoylated [47]—for instance, AMPA receptors [48,49,50,51], NMDA receptors [52,53,54,55], large conductance calcium-activated potassium channels [56], dopamine transporters [57], synaptic vesicle fusion machinery proteins [58], voltage-gated ion channels [59], kinases [60], and G-protein coupled receptors [61]. More recently, the high throughput mass spectrometry approach has provided data on brain palmitoylome [62,63,64,65]. Considering hundreds of synaptic proteins identified in mass spectrometry studies, we are just beginning to unravel the realm of lipid PTMs. Well-established [66,67] and recently developed methods now allow in silico prediction of S-palmitoylation sites in mammalian protein datasets [65]. In addition, most recently, Wild and coworkers curated and analyzed expression data for the proteins that regulate S-palmitoylation from publicly available RNAseq datasets, providing a comprehensive overview of their distribution in the mouse nervous system [68].

### 1.3. Depalmitoylation Enzymes

Unlike palmitoylating enzymes, relatively less is known about the localization and structure of the depalmitoylating enzymes. Deacylation is catalyzed by palmitoyl-protein thioesterases (PPTs). The first depalmitoylation enzyme, palmitoyl-protein thioesterase 1 (PPT1), was described in 1993 [69] and was shown to be present in the synaptosomes and lysosomes [70,71]. PPT2, a homolog of PPT1, remains expressed primarily in the skeletal muscle [72], unlike PPT1, which is highly expressed in the brain. The search for the thioesterases responsible for the deacylation of α subunits of heterotrimeric G proteins (guanine nucleotide-binding protein G(s) subunit alpha isoform) resulted in the description of the first cytosolic depalmitoylation enzyme—acyl-protein thioesterase (APT1) [73]. APT2 shares a high degree of similarity with APT1 [74], and both APT1 and APT2 contain unusual α/β serine hydrolase fold but differ in the substrate specificity [74,75]. The third acyl-protein thioesterase, APT1-like (APT1L) protein, was revealed using a bioinformatic approach [76]. In an attempt to identify enzymes responsible for the depalmitoylation of PSD-95, depalmitoylases alpha/beta hydrolase domain-containing protein 17 (ABHD17) isoforms A/B/C were discovered [77]. The most recently identified depalmitoylating enzyme is ABHD10—a mitochondrial protein with S-depalmitoylase activity [78]. All these enzymes belong to the metabolic serine hydrolases superfamily, with a conserved α/β-hydrolase domain [79].

### 1.4. Pharmacological Manipulation of Protein Palmitoylation

The methods of pharmacological manipulation of palmitoylation machinery have been recently reviewed [80,81], showing two main groups of compounds: lipid-competitive inhibitors and non-lipid-competitive inhibitors. The first group of compounds inhibits ZDHHC activity: 2-bromopalmitate (2-BP), cerulenin, and tunicamycin. 2-BP is the palmitate analog and the most commonly used inhibitor of palmitoylation. However, while it is an irreversible nonselective inhibitor of PATs, which blocks palmitate addition [82], it has poor specificity and affects many targets beyond PATs [83]. For instance, 2-BP was shown to inhibit protein deacylation [84] via inhibiting thioesterases APT1 and APT2 [85]. Cerulenin and tunicamycin are antibiotic compounds known to inhibit palmitoylation [86]. Recently, a new tool to study palmitoylation emerged with the development of acrylamide-based, cyano-myracrylamide (CMA), which exhibits lower toxicity and does not inhibit deacylating enzymes [87]. In addition, in a high-throughput screening study, five chemotypes of non-lipid competitive inhibitors were identified [88]. One of them, 2-(2-hydroxy-5-nitro-benzylidene)-benzo[b]thiophen-3-one (named compound V, CV) inhibited the activity of human ZDHHC2 and ZDHHC9, but in contrast to 2-BP, the inhibition was reversible [89]. Other inhibitors include curcumin, which blocks autoacylation of the ZDHHC3 [90], and artemisinin, which is known to inhibit the enzymatic activity of ZDHHC6 [91].

A widely used inhibitor of depalmitoylation enzymes is palmostatin B, which inhibits APT1 and APT2 and serine hydrolases from the ABHD family [21]. It was developed in 2010 as one of the new compounds inhibiting APT1. Among palmostatin A–D, palmostatin B proved to be the most potent and was shown to disturb the depalmitoylation of Ras protein [92]. Later, in another study, palmostatin M was discovered in the search for new inhibitors of acyl-protein thioesterases and proved to be more active than its predecessor [93]. There are also other inhibitors in use which are selective towards the isoforms of the APT enzyme—namely, APT1 inhibitor [94], APT2 inhibitor, and ML349 [95]. In contrast to the above mentioned inhibitors, N-(tert-Butyl) hydroxylamine (NtBuHA), a nontoxic hydroxylamine derivative, is able to cleave thioester linkage, mimicking the action of thioesterases [96]. Figure 1 illustrates palmitoylation-depalmitoylation cycles with key enzymes and their inhibitors.

## 2. Palmitoylation—The Roles in the Function and Plasticity of Synapses

The evidence for the neuronal activity influencing palmitoylation of synaptic proteins is increasing. Reciprocally, manipulation of the palmitoylation dynamics has an impact on the efficacy of synaptic connections between neurons. A common approach to study the role of protein palmitoylation in the function of synapses is the combination of pharmacological treatments and genetic engineering with electrophysiological recordings or live imaging. In this chapter, we discuss some of the experimental evidence linking synaptic plasticity and protein palmitoylation. We also review recent insights on the role of palmitoylation in the function of excitatory and inhibitory synapses, along with synaptic plasticity (Figure 2). The roles of palmitoylation in structural plasticity [97], dendritic spine remodeling [98], and the development and plasticity of neuronal connections [99], have been recently reviewed elsewhere.

### 2.1. Synaptic Transmission and Synaptic Plasticity

Neurons need to communicate with each other in order to efficiently propagate information through the neuronal networks. In the central nervous system of mammals, glutamate and γ-aminobutyric acid (GABA) are the most common excitatory and inhibitory neurotransmitters [100]. AMPAR (α-amino-3-hydroxy-5-methyl-4-isoxazolepropionic acid receptor) and NMDAR (N-methyl-D-aspartate receptor) are ligand-gated ion channels whose activation causes an inward current of positive ions, depolarization of the postsynaptic membrane, and the generation of excitatory postsynaptic potential (EPSP) [101]. Activation of GABA_A_Rs (GABA type A receptors) causes an inflow of chloride ions, causing the hyperpolarization of the postsynaptic membrane and the generation of inhibitory postsynaptic potential (IPSP) [102]. Information sent to postsynaptic neurons (synaptic transmission) becomes “uncoded” as a result of the summation of EPSP and IPSP. The “propagation” of information occurs when the postsynaptic neuron exceeds the threshold for firing action potential [103].

Learning and memory require synaptic plasticity alteration through the number and strength of existing synaptic connections. Memory traces may also be encoded by use-dependent modification of synapses [104]. Likewise, chemical synapses do not act as only static relays of information, as the effectiveness with which they perform synaptic transmission could be changed. Thus, it is evident that such synaptic plasticity may take various forms [105]. Similarly, functional plasticity is manifested by activity-dependent modification of the efficacy of synaptic transmission at already existing synapses. These changes may be either short- or long-lasting. Most forms of short-term synaptic plasticity are caused by the activity-dependent accumulation of Ca^2+^ ions (residual Ca^2+^) in the presynaptic bouton [105,106]. In addition, short-term plasticity acts as a temporary storage buffer of new information that may further undergo either a process of consolidation towards long-term memory or expiration. In experimental conditions, the extensively studied forms of synaptic plasticity have been long-term potentiation (LTP) and long-term depression (LTD) of synaptic transmission in various circuits [104,107,108]. The first observation of LTP was reported in the dentate area of the rabbit hippocampus [109]. However, since then, the synapses formed by the Shaffer collaterals on the dendrites of CA1 pyramidal neurons have become the model excitatory synapses in functional studies [110]. The hippocampus is a structure involved in the processes of learning and memory formation [111,112,113,114], and it became the principal model for studying plasticity. The molecular mechanisms behind LTP and LTD, which are triggered by the activation of NMDARs and the role of AMPARs in that process, have been well described [115,116,117,118]. Homeostatic plasticity is another form of response of the synaptic circuits that counters the potential effects of long-term synapse-specific plasticity by globally affecting the transmission through all synapses on a given neuron [119,120]. Thus, average neuronal activity levels are maintained by a set of mechanisms that dynamically adjust synaptic strengths to promote stability [119]. In experimental conditions, this may be observed upon chronic blockade of AMPARs or neuronal activity with tetrodotoxin (TTX), which leads to scaling up of excitatory synaptic weights [119]. Relatively recently, plasticity of inhibitory synaptic transmission has been described (see [121] for a review). The strength of inhibitory synapses can be regulated by changes in their presynaptic [122] or postsynaptic sites [123]. There are also two types of inhibitory long-term plasticity—iLTP (GABAergic inhibitory long-term potentiation) and iLTD (inhibitory long-term depression). Similarly to glutamatergic synapses, the magnitude of GABAergic postsynaptic plasticity largely depends on the changes in the number of receptors available at the plasma membrane and is supported by scaffolding proteins [123,124,125].

Both LTP and LTD are correlated with learning and memory [126,127,128]. Memory formation requires similar functional and molecular synaptic modifications as those previously found to accompany long-term plasticity phenomena experimentally induced with patterned stimulations [129,130]. Thus, behavioral training leading to synaptic potentiation in vivo and LTP evoked by high-frequency electrical stimulation occlude each other. This is a strong argument indicating common mechanisms of both phenomena [130]. Molecular pathway studies of LTP and LTD may contribute to better understanding of experience-dependent development, addiction, and neurological disorders, such as Alzheimer’s disease and Parkinson’s disease (reviewed in [105]). It is clear, however, that no patterned exogenous stimulation can reproduce a complex pattern of endogenous activity of neuronal networks occurring in vivo. Changes solely in the synaptic strength of synaptic connections, such as LTP alone, are insufficient to explain memory formation [131,132]. Indeed, neurons may significantly enhance the information storage capacity and learning by scaling dendritic and somatic excitability [133,134,135] (for review, see [136]). Thus, alterations in synaptic strength and neuronal excitability expressed in firing rate, firing threshold, or gain can underlie memory storage and information processing.

### 2.2. Neuronal Activity and Synaptic Plasticity Affect Palmitoylation/Depalmitoylation Cycles

S-palmitoylation is often considered the only type of lipid modification that is reversible and thus may dynamically regulate the state of proteins. Early research on S-palmitoylation dynamics in non-neuronal cells indicated that palmitoylation/depalmitoylation cycles may range from seconds [137] to 20–50 min [138,139,140]. This fast rate of the palmitoylation/depalmitoylation cycling allows it to act as a dynamic switch regulating the fate of protein, after activation of signaling pathways in neurons during synaptic plasticity. S-palmitoylation may play a similar role to other PTMs with rapid turnover rates (i.e., phosphorylation) in controlling localization and the functions of the proteins. Proteomic data analysis indicates that nearly 10% of the human genome encodes protein that can be palmitoylated, out of which, 41% are synaptic proteins [63]. Although not all proteins are dynamically palmitoylated [62], it is quite evident that lipidation of synaptic proteins may have a significant impact on their function. Indeed, several synaptic proteins undergo palmitoylation or depalmitoylation in neurons almost immediately following a period of enhanced neuronal activity. In a seminal study by Kang and coworkers, brief enhancement of the activity of cortical neurons with glutamate resulted in a rapid and protein-selective depalmitoylation of PSD-95, Ras, and cell division control protein 42 homolog (Cdc42), among many others [43]. In accordance with that, strong depolarization of neuronal membranes with KCl for 5 min was shown to lead to rapid depalmitoylation of PSD-95 [31]. This process was found to be ZDHHC2-dependent and involve PSD-95 translocation away from the postsynaptic membrane, as visualized with a PSD-95 palmitoylation-specific biosensor (conformation-specific recombinant antibody). By application of 2-BP, which blocks the newly occurring palmitoylation, it was found that in basal conditions, almost the entire population of palmitoylated PSD-95 (90%) in young primary neuronal culture, unlike in older cultures, participated in the dynamic palmitoylation cycles. This indicates that depalmitoylation kinetics of PSD-95 are more dynamic (plastic) in the early stages of synaptic development and maybe decelerate as synapses mature. This is true only for PSD-95, since palmitate on the subunits of AMPA and NMDA receptors (GluA1, GluN2A), metabotropic glutamate receptor 5 (mGluR5), G-protein subunit Gq, and GTPase HRas hardly turned over in neurons within 6 h post 2-BP treatment [141]. Another study found that stimulation of neurons with glutamate and an NMDAR agonist, glycine, for 2 min, resulted in a rapid reduction of available PSD-95 in the synapses 15 min later [142]. Inversely, when neuronal activity is reduced or silenced, the promotion of synaptic proteins should occur. Indeed, TTX applied for at least 2 h was shown to enhance PSD-95 palmitoylation, recruiting this protein to postsynaptic density and facilitating the clustering of postsynaptic receptors [31].

One commonly used in vitro model of inducing neuronal plasticity involves glycine/bicuculline, which chemically promotes synaptic LTP by selective activation of synaptic NMDARs [26,143]. In contrast to chemical LTP, chemical LTD induced with glycine/NMDA activates both synaptic and extrasynaptic NMDARs, resulting in AMPAR internalization and synaptic depression [144,145]. Primary hippocampal cultures placed in a conventional bathing solution lacking glycine and Mg^2+^ showed selective activation of synaptic NMDARs by brief bath application of NMDAR co-agonist glycine [143]. This led to the rapid cell surface expression of AMPARs and upregulation of miniature excitatory postsynaptic currents (mEPSCs), amplitude, and frequency. In contrast, blockade of synaptically active NMDARs with MK-801, followed by the application of NMDA and glycine, resulted in the activation of extrasynaptic NMDARs, and as a result, synaptic LTD [143]. In that context, an increase in the δ-catenin palmitoylation mediated by ZDHHC5 was observed as soon as 20 min post-induction of increased neuronal activity with a glycine cLTP protocol but not cLTD [26]. Similarly, in excitatory synapses of in vitro neuronal cultures, plasticity-related gene 1 protein (PRG-1), palmitoylation was elevated as early as 10 min post cLTP induction [41]. Altogether, this evidence suggests that palmitoylation of some synaptic proteins may be highly dynamic and therefore relevant to neuronal activity-dependent induction and maintenance of synaptic plasticity. Moreover, a picture emerges that acute depolarization of neurons, which promotes their enhanced activity, leads to preferential depalmitoylation of synaptic proteins, yet silencing of neuronal activity generally (but not always) leads to the reverse process.

Neuronal activity also seems to affect palmitoylation status in a protein-specific manner. The already mentioned glutamate treatment that was shown to downregulate the palmitoylation levels of some key synaptic proteins, such as PSD-95, did not affect the palmitoylation of some other proteins, such as GluR2 or synaptosomal-associated protein 25 (SNAP25), and lead to increased palmitoylation levels of another subset of proteins, namely, calnexin and secretory carrier membrane protein 1 (Scamp1) [43]. Another study found that a glutamate receptor interacting protein-1b (GRIP1b), known to be palmitoylated by ZDHHC5 and ZDHHC8 [29], does not change its palmitoylation status in response to synaptic activity [99]. Yet another set of example comes from a study where cLTP induction did not affect the abundance of the palmitoylated form of proteins implicated in synaptic plasticity, such as calcium/calmodulin-dependent protein kinase type II subunit alpha (CaMKIIα), G-protein subunit α11/q, and others [41]. In contrast, intraperitoneal injection of kainic acid, which induces convulsive seizures, resulted in upregulation or no change in the palmitoylation status of individual proteins after 30 min, and protein depalmitoylation was not at all observed. Altogether, palmitoylation/depalmitoylation of synaptic proteins seems to have a unique temporal profile for each protein, besides remaining protein-specific. However, the general rules linking the pattern of neuronal activity, chemical cues, and intracellular cascades leading to protein-specific palmitoylation status change, remain largely unexplored.

The current literature provides evidence that neuronal activity affects the subcellular localization of PATs. ZDHHC5 was shown to be in the vicinity of dendritic spines, and in response to changes in neuronal activity, was rapidly recruited to modulate the palmitoylation state of synaptic proteins. Additionally, ZDHHC5 can be bound to PSD-95 and Fyn kinase, stabilizing at the synaptic membrane, thereby supporting synaptic plasticity [26]. Induction of cLTP with glycine in the neuronal cultures resulted in increased abundance of ZDHHC5 in the excitatory synapses, whereas post cLTD treatment, neither the excitatory synapses nor the inhibitory synapses exhibited any changes in ZDHHC5 abundance [26]. During cLTP, ZDHHC5/PSD-95/Fyn kinase complex was disrupted and ZDHHC5 was translocated to dendritic shafts, where it palmitoylated δ-catenin. Though this translocation of ZDHHC5 was dynamic, its surface levels were initially found to be decreased and then gradually increased in the 3–20 min post-stimulation time window [26]. This showed that ZDHHC5 may rapidly translocate in response to the induction of synaptic plasticity. ZDHHC5 may also support this translocation process by palmitoylating δ-catenin and promoting AMPARs surface insertion, which is crucial for synaptic efficacy enhancement [26]. Another key palmitoyltransferase, ZDHHC2, was shown to be distributed in the dendrites and cell body in small vesicular-like structures [27]. In vitro neuronal activity blockade with TTX for 2–12 h promoted palmitoylation of PSD-95 and its accumulation, in addition to increasing the AMPARs clustering in the excitatory synapses [27]. Knockdown of ZDHHC2 blocked this process completely, and it was found that the observed changes to PSD-95 and AMPARs are not associated with increased ZDHHC2 activity but rather to the distinct localization of the enzyme. Indeed, ZDHHC2 senses changes in synaptic activity and becomes rapidly translocated from dendritic shafts toward postsynaptic membranes. Thus, a picture emerges: that individual ZDHHC members may be dynamically recruited to active excitatory synapses to support synaptic plasticity. Moreover, dynamic changes in neuronal activity may differentially affect the localization of elements of the enzymatic machinery required for S-palmitoylation.

Palmitoylation of synaptic proteins may change dynamically upon synaptic plasticity. AMPARs and their associated scaffold proteins play key roles in the function and plasticity of excitatory synapses in the central nervous system [146]. In addition to the palmitoylation of the AMPARs themselves, palmitoylation of synaptic proteins that directly or indirectly interact with AMPARs, such as PSD-95, glutamate receptor-interacting protein 1 (GRIP1) glutamate receptor-interacting protein 2 (GRIP2), protein interacting with C kinase-1 (PICK1), and A-kinase anchoring protein 79/150 (AKAP79/150), can also be an important regulatory factor for AMPAR trafficking and function [47,147,148]. Palmitoylation of AKAP79/150 regulates NMDAR-dependent excitatory synaptic potentiation [149]. Changes in neuronal activity affect the palmitoylation levels of AKAP79/150. LTP modeled in mice hippocampal slices with either glycine or high-frequency stimulation of afferent fibers (HFS) led to an increase in AKAP79/150 palmitoylation levels 30 min after induction and remained elevated an hour later [150]. Additionally, post HFS, there was an upsurge in surface expression of AKAP79/150 and its association with PSD-95, suggesting palmitoylation’s role in the protein’s translocation to the proximity of the synaptic membrane. Moreover, the HFS-evoked enhancement of AKAP79/150 palmitoylation and its interaction with PSD-95 in hippocampal slices was prevented by pre-treatment with a palmitoylation inhibitor 2-BP, further supporting palmitoylation’s involvement in activity-related changes to AKAP79/150 localization and function. Protein kinase A (PKA) is another important interactor of AKAP79/150, and together, they influence AMPAR function at the spine surface [151]. HFS promotes synaptic expression of PKA and its association with AKAP79/150, and both are abolished by 2-BP, which strongly suggests palmitoylation’s involvement in these processes [150]. AKAP79/150 is also palmitoylated at its N-terminal by ZDHHC2. HFS does not influence ZDHHC2 expression, but it leads to increased palmitoylation of the PAT. As previously mentioned, ZDHHC2 localizes mainly to the recycling endosomes, and upon stimulation, changes its localization to a postsynaptic density, where it can interact with AKAP79/150 and regulate palmitoylation of PSD-95 [150]. Altogether, during LTP, palmitoylation may play an important role in facilitating the interaction between PKA and AKAP79/150.

Dynamic neuronal activity is essential for the encoding of information in neural networks. Subcellular localization or expression of ZDHHC members may change with neuronal activity. For instance, pathological neuronal activity observed in the temporal neocortex of intractable epilepsy patients and in the hippocampus and cortex of rats with experimentally induced epilepsy was linked with neuron-specific downregulation of ZDHHC3 mRNA and protein [152]. Membrane surface trafficking and clustering of AMPARs, NMDARs, and GABARs are regulated through ZDHHC3-mediated palmitoylation [153,154]. Therefore, it has been speculated that the decrease in ZDHHC3 may have a detrimental effect on the function of GABAergic synapses, promoting NMDARs and AMPARs localization on the neuronal surface, and in this way contribute to epilepsy [152].

Some experiments further support the idea that engagement of palmitoylation machinery is supportive to homeostatic plasticity. Both acute (2 h) and prolonged (24–48 h) downregulation of neuronal activity with TTX or with an AMPAR blocker kynurenic acid promoted the recruitment of ZDHHC2 to the postsynaptic density, where it increasingly palmitoylated its substrate, PSD-95, and promoted PSD-95’s association with the postsynaptic density and its ability to cluster postsynaptic glutamate receptors [27]. In contrast, prolonged elevation of synaptic activity by GABA_A_R antagonist bicuculline (48 h) reduced PSD-95 palmitoylation. Altogether, AMPARs in the synapses are either targeted to or dispersed from synapses, depending on the PSD-95 palmitoylation state, suggesting that a dynamic cycle within a spine modulates bidirectional homeostatic AMPAR plasticity [155]. With regard to NMDARs, extended treatment with TTX increased palmitoylation of GluN2A and GluN2B [154]; however, no functional studies were performed, and it is not very clear whether NMDAR-mediated synaptic current may also be altered in these circumstances. Another example comes from a study on an AMPAR-interacting transmembrane protein, synapse-differentiation-inducing 1 protein (SynDIG1), showing its synaptic localization in the neuronal cultures and its palmitoylation status in organotypic hippocampal slice cultures being upregulated post-TTX treatment [156,157]. While the direct impact of palmitoylated SynDIG1 on synaptic function was not studied, it may be possible that palmitoylation-supported synaptic localization of SynDIG1 is an additional mechanism reinforcing AMPAR recruitment to the postsynaptic membrane during homeostatic plasticity.

### 2.3. Impact of Protein Palmitoylation on Synaptic Transmission and Various Forms of Neuronal Plasticity

#### 2.3.1. Pharmacological Modifications of Palmitoylation Machinery

NtBuHA treatment of acute brain slices, leading to a global palmitoylation-deficient state, selectively reduced the amplitude of AMPAR-mediated mEPSCs in the hippocampal neurons [158]. It was also shown that NtBuHA in slices depalmitoylated PSD-95, AKAP150, and AMPAR and NMDAR subunits, and led to a reduction in the interaction of PSD-95 with AMPARs and reduced surface expression of GluA1/2 subunits of AMPARs [158]. A lipid-competitive inhibitor 2-BP applied to neuronal cultures for 8 h reduced the amplitude and frequency of mEPSC and decreased the AMPAR/NMDAR ratio by disrupting the clusterization of PSD-95 and diminished AMPARs’ availability [159]. These findings support a view that palmitoylation of synaptic proteins is required for basal excitatory synaptic transmission, and both the rapid depalmitoylation induced by NtBuHA and the disruption of the palmitoylation/depalmitoylation balance with 2-BP directly interfere with the efficacy of electrical synaptic signaling. However, it has also been reported that 24 h incubation of acute brain slices with 2-BP did not affect the mEPSCs’ frequency, amplitude, or AMPAR-mediated current density recorded in CA1 hippocampal region in acute brain slices [160]. This discrepancy in observations may be due to different concentrations of 2-BP and the model used in both studies as differences in inhibitor efficacy in tissue versus culture models may exist.

Inhibition of palmitoylation with 2-BP in hippocampal slices impaired LTP in Sch-CA1 excitatory synapses [161]. In contrast to this study, two other studies did not observe the effect of 2-BP on hippocampal HFS-LTP [160,162]. In addition, manipulation of APT1 activity with ML383, which is expected to promote palmitoylation, did not affect HFS-LTP in the hippocampus [161]. Interestingly, a high-fat diet promoting palmitate abundance has a negative impact on the magnitude of hippocampal LTP, but 2-BP ameliorated this effect [160]. 2-BP also ameliorated LTP impairment in the offspring of the maternal high-fat diet animal group [162]. Altogether, a picture emerges that in neuronal tissue, pharmacological manipulations of palmitoylation machinery which lead to either up- or downregulation of palmitoylation, have little impact on long-term synaptic plasticity. However, the use of 2-BP may reverse plasticity impairment associated with higher palmitate levels resulting from a high-fat diet. In acute brain slices, the global depalmitoylation of brain palmitoylome has been shown to occur following NtBuHA treatment [158]. However, there is virtually no evidence that 2-BP is effectively blocking palmitoylation in brain slices. Thus, while 2-BP has documented effects in vitro, there is hardly any evidence for the impact of this drug on excitatory synaptic transmission in nervous tissue, unlike NtBuHA. The reason that underlies these apparent discrepancies remains unknown. However, it should be noted that the mechanism of action of both the drugs is different. NtBuHA cleaves all existing thioester bonds directly, leading to depalmitoylation of all proteins, and 2-BP inhibits de novo palmitoylation. Thus, under conditions of low PAT activity, the effect of 2-BP would be less extensive compared to NtBuHA, and it may be pronounced in pathological conditions.

While it is clear that palmitoylation of synaptic proteins supports LTP, the evidence for its role in synaptic LTD in pharmacological studies is scarce. In the cultures of cerebellar Purkinje cells, LTD of EPSCs can be studied by pairing iontophoretic glutamate pulses applied to cells with membrane depolarization. That type of LTD requires PICK1 and is abolished upon application of palmitoylation inhibitor 2-BP [149,163]. PICK1 binds and regulates GluA2-containing AMPARs. It was also shown that ZDHHC8 but not ZDHHC5 was required for the development of LTD and that ZDHHC8 palmitoylated PICK1 at C414 [149]. Thus, considering this one exemplary study, ZDHHC activity may support not only LTP but also LTD expression.

Pharmacological manipulation of protein palmitoylation was shown to effectively block homeostatic plasticity, which is defined broadly as a set of neuronal changes that restore activity to a setpoint following perturbation [119,120]. For instance, increases in the miniature EPSCs’ amplitude, levels of GluA1/A2 AMPAR subunits, and palmitoylation levels of PSD-95 and GluN2A/2B associated with homoeostatic plasticity in cultured hippocampal neurons were reversed by palmitoylation inhibitor 2-BP. In addition, NtBuHA prevented the enhanced surface expression of GluA subunits, and eventually, enhancement of PSD-95 and GluN subunits palmitoylation [161]. Another insight into the potential impact of palmitoylation on synaptic scaling in homeostatic plasticity comes from an experiment where inhibition of depalmitoylation activity of APT1 was achieved by lentivirus-delivered gene silencing or by the use of the inhibitor ML348. In this model of more pronounced palmitoylation, increased mEPSC, increased AMPAR/NMDAR ratio, and increased presynaptic release were observed [161]. Interestingly, lentivirus silencing of another thioesterase PPT1 did not result in changes in mEPSC, indicating that depalmitoylation by APT1 rather than PPT1 plays a crucial role in homeostatic scaling of excitatory synapses. Indeed, APT1, unlike PPT1, may be responsible for controlling the palmitoylation status of many synaptic proteins, including AMPAR, NMDAR, PSD-95, and stargazin; and determining surface availability of AMPAR subunits GluA1/2 [161]. Altogether, silencing of neuronal activity or promotion of PSD-95/AMPARs/NMDARs’ palmitoylation leads to synaptic strengthening at excitatory synapses, suggesting that homeostatic synaptic scaling requires protein palmitoylation.

#### 2.3.2. Cysteine Mutants and Protein Engineering

With the advent of genetic engineering, studying recombinant proteins and transgenic animals has advanced the understanding of protein palmitoylation. Cysteine residue is required for the formation of thioester bonds with palmitate. Thus, the substitution of cysteines in recombinant proteins has been used to provide insights into the role of palmitoylation in the function of the synaptic proteins. The consequences of disturbed palmitoylation of the GluA1 subunit were verified in several studies. GluA1 C-terminal palmitoylation-deficient (GluA1C811S) mice exhibited no change in the excitatory to inhibitory synaptic currents ratio in dentate gyrus granule neurons, indicating no gross alteration in the excitatory/inhibitory balance [164]. In addition, mutants did not differ in hippocampal HFS-LTP and NMDA-induced LTD [164], indicating that C-terminal palmitoylation of GluA1 subunits is not required for these forms of plasticity. However, mutant mice exhibited excessive excitability in response to seizure-inducing stimulation. These mice also exhibited reduced anticonvulsive effects in response to anticonvulsants blocking sodium and calcium channels and suppressing excess excitation [165]. Thus, palmitoylation of the GluA1 at C811 may not be important in basal synaptic activity or plasticity. However, it may regulate the threshold of seizure occurrence upon seizure-inducing activity. The functional assembly of wild-type and cysteine mutants of AMPARs was also examined in transfected HEK 293T cells. Electrophysiological recordings in cells expressing both GluR1C585S/GluR2C610S and GluR1C811S/GluR2C836S mutants indicated similar current–voltage relationships, suggesting that cysteine mutations do not disrupt the functional heteromeric assembly of AMPARs in heterologous expression systems, nor modify AMPARs’ conductance [154]. Thus, it is ascertained that palmitoylation of AMPARs does not directly affect the currents gated by these receptors.

GluN1 subunits of the NMDAR are not palmitoylated; however, two conserved clusters of cysteines, cluster I and II, have been described in both GluN2A and GluN2B subunits of NMDARs. Palmitoylation of cysteines of cluster I belonging to the GluN2 subunit increases, whereas the same process in cluster II decreases the surface expression of NMDARs [166]. The impact of palmitoylation of NMDAR subunits on functional synaptic incorporation of NMDARs was investigated with the use of cysteine mutants of GluN2 subunit [167]. The cysteine substitution with non-palmitoylatable serines in cluster I resulted in a decreased membrane incorporation index of NMDARs and reduced evoked EPSCs in CA1 neurons. In contrast, protein mutants with substituted cysteine residues in cluster II had a normal incorporation index. Moreover, palmitoylation-deficient mutants in cluster I exhibited a tendency for reduced AMPAR and NMDAR-mediated synaptic currents [167]. In the most recent study, the gating properties of a recombinant NMDAR mutant with C849, C854, and C871 were investigated in HEK293 cells [168]. It was found that a lack of these palmitoylatable cysteines reduced the probability of recombinant NMDAR opening by 50%, and open-time intervals were shorter. If this would hold in physiological conditions, palmitoylation of the cysteine cluster located at the carboxy-terminal domain of GluN2B subunit could regulate the gating of NMDARs, and thus the Ca^2+^ current critical for synaptic plasticity and cell signaling in general. Altogether, studies of recombinant AMPA and NMDA receptors suggest the existence of hot spots where palmitoylation could affect either the gating properties of the receptors or otherwise functionally regulate their membrane surface availability. It is noteworthy that the rise in intracellular Ca^2+^ initiated depalmitoylation of NMDARs [168]. Thus, it is plausible that NMDAR activity via Ca^2+^ could indirectly modulate the palmitoylation state of the receptor. Furthermore, the sensitivity of native NMDARs and recombinant GluN1/GluN2B receptors to the endogenous inhibitory neurosteroid 20-oxo-5β-pregnan-3α-yl 3-sulfate (PAS) and its synthetic analogs are increased following receptor stimulation in the presence of Ca^2+^ ions. This process is modulated by palmitoylation, as it was found that neurosteroid-induced reduction of NMDAR-mediated currents was more pronounced when tunicamycin, an inhibitor of palmitoylation, was applied [168].

Mounting evidence suggests that palmitoylation of PSD-95 may regulate the synaptic function and plasticity of excitatory synapses by adjusting AMPAR and NMDAR membrane availability. Overexpression of PSD-95 results in increased amplitude and frequency of mEPSCs, thereby mimicking LTP. It also converts silent synapses to functional synapses. In addition, LTP is completely occluded in cells expressing PSD-95, whereas LTD is greatly enhanced, suggesting that the control of synaptic AMPARs by PSD-95 and synaptic plasticity share common mechanisms [169]. Suppression of synaptic activity or reducing extracellular Ca^2+^ results in the accumulation of PSD-95 clusters in neuronal cultures. This was not observed upon treatment with 2-BP, suggesting that PSD-95 palmitoylation remains crucial in this process. Indeed, PSD-95 is palmitoylated at two cysteine residues (C3 and C5) at its N-terminus, and palmitoylation at this locus is required for clustering of PSD-95 [159]. In addition, neurons expressing C3 and the C5 PSD-95 mutant exhibit impaired AMPAR surface availability and reduced EPSCs [170]. The mechanism of this process has been recently described. Inward Ca^2+^ flux increases Ca^2+^/calmodulin (CaM) binding to the N-terminal domain of PSD-95. The N-terminal capping of PSD-95 by CaM blocked palmitoylation of C3 and C5, which is required for postsynaptic PSD-95 targeting and the binding of cyclin-dependent kinase-like 5 (CDKL5), a kinase important for synapse stability. Subsequently, dissociation of PSD-95 from the postsynaptic membrane was observed [142]. The impact of PSD-95 palmitoylation primarily on the AMPAR-mediated signals, and not NMDARs, may be explained by PSD-95 conformation. In a study in HEK293 cells, PSD-95 was found in the compact conformation when depalmitoylated [171]. In the same expression system, it was found that PSD-95 palmitoylation is more dynamic when PSD-95 is associated with AMPARs and more stable when it is associated with NMDARs. This suggests that PSD-95 palmitoylation cycling would be dynamically driven in the AMPAR nanodomain but slower in the NMDAR nanodomain [171]. The palmitoylation state of PSD-95 may separate nanodomains, increase the packing density of the receptors, and thereby increase the density of local current at AMPARs and Ca^2+^ current density at NMDARs. In addition, it may allow strategic positioning of the clusters of NMDARs closer to or further away from intracellular, downstream, calcium-dependent effectors to regulate the effects of the intracellular calcium influx.

ZDHHC2, ZDHHC3, ZDHHC5, ZDHHC7, ZDHHC8, and ZDHHC15 are involved in the palmitoylation of PSD-95 at cysteine 3 and 5 (reviewed in [51]). While PSD-95 may be constitutively palmitoylated by ZDHHC3, palmitoylation in response to synaptic activity is mediated by ZDHHC2, whereas PSD-95 clustering in neurons is regulated by ZDHHC17 [32]. Fukata and coworkers identified the members of the ABHD17 serine hydrolases as the PSD-95 depalmitoylating enzymes [141]. In addition, ABHD17 was found to be selective for PSD-95 in neurons compared to other tested proteins, i.e., GluA1/2 subunits of AMPARs, the GluN2A subunit of NMDARs, the γ2 subunit of the GABA_A_ receptor, Gα, HRas, and vGLuT1. ABHD17B-mediated PSD-95 depalmitoylation is another mechanism that allows the reduction of synaptic clusters of PSD-95 and AMPA receptors [31]. Altogether, palmitate cycling on PSD-95 seems to be crucial for AMPARs’ surface availability, and thus, this process may regulate synaptic strength and activity-dependent plasticity. While PSD-95 is a substrate for many enzymes of palmitoylation machinery, ZDHHC2 seems to be the major PAT involved in the activity-regulated palmitoylation of PSD-95.

AKAP150 scaffolds kinases and phosphatases to regulate GluA1 phosphorylation and trafficking, and trafficking of AKAP150 itself is modulated by palmitoylation on two cysteine residues. Palmitoylation-deficient AKAP150 C36,123S (AKAPCS) knock-in mutant mice were recently developed [172], having no changes in dendritic spine numbers or morphology in CA1 stratum radiatum of ex vivo brain slices compared to the wild types. However, on the functional level, excitatory synaptic transmission in AKAPCS mice and cultured neurons revealed larger mEPSCs amplitudes. In agreement with that, an increased AMPAR/NMDAR ratio of evoked synaptic currents was observed in synapses of CA1 pyramidal neurons. Palmitoylation of AKAP150 is crucial for synaptic plasticity, as AKAPCS mutant mice exhibited impaired HFS-LTP while the magnitude of low frequency induced LTD (LFS-LTD) remained unchanged. The AKAPCS mutant did not exhibit altered presynaptic release probability. Thus, LTP deficits in AKAPCS mice were specifically related to impaired regulation of AMPARs’ permeability to Ca^2+^ ions [172]. It also showed the importance of palmitoylation of the postsynaptic scaffolding molecule AKAP150 in AMPAR’s synaptic subunit composition in the context of LTP. These results may be partially explained by a recently discovered fact that palmitoylation determines the nanoscale localization and mobility of AKAP150 within a postsynaptic compartment [173].

Another example of the impact of palmitoylation on the function of excitatory transmission comes from a study of AMPA receptor binding protein (ABP), a multi-PDZ domain scaffold that binds and stabilizes GluR2/3 subunits of AMPARs at synapses. The palmitoylated N-terminal splice variant of ABP (pABP-L) is concentrated in spine heads, whereas a non-palmitoylated form is located intracellularly [174]. Postsynaptic Sindbis (virus) expression of pABP-L in hippocampal primary cultures increased the spine density and synaptophysin puncta, indicating an increased number of synaptic contacts [175]. In addition, pABP-L overexpression increased AMPAR-mediated mEPSC amplitude and frequency and elevated surface levels of GluR1 and GluR2. In contrast, a non-palmitoylated pABP-L mutant (C11A) did not change spine density or size. It should be noted that both ABP and its closely related protein, GRIP, are synthesized in a palmitoylated form that is targeted to the spine and a non-palmitoylated form that is targeted to intracellular clusters [174,176]. Palmitoylation by ZDHHC5/8 was shown to target GRIP1 at dendritic endosomes for regulating AMPAR trafficking [29]. Since ABP/GRIP colocalized with cadherin and δ-catenin at excitatory synapses, it has been suggested that palmitoylation of ABP may control synaptic function and synaptic plasticity [175]. As mentioned earlier, palmitoylation of PICK1 protein plays a role in AMPAR regulation in LTD. PICK1 and ABP/GRIP both bind to the C terminus of the GluR2 subunit of AMPARs, and the interaction of PICK1 with ABP/GRIP is a critical step in controlling GluR2 trafficking [177]. Cerebellar Purkinje neurons transfected with PICK1 protein cysteine mutant exhibited a lack of LTD when treated with either iontophoretic glutamate pulses in conjunction with depolarization or application of protein kinase C activator PDA. The results obtained with this approach corroborate the findings with the use of palmitoylation inhibitor 2-BP [148]. Altogether, this indicates that palmitoylation of ABP/GRIP and PICK1 proteins may control AMPAR targeting and trafficking, which is central to synaptic transmission and plasticity.

Activity-regulated cytoskeletal-associated protein (Arc or Arg3.1) is induced in the neurons in response to the neural activity and is crucial for many forms of synaptic plasticity and memory [178]. Arc influences synapse function via multiple pathways, including interaction with F-actin and regulation of AMPAR trafficking and internalization and transcription of the AMPAR subunits (reviewed in [179]). A part of the Arc protein pool was found to be present in membrane rafts of mouse synaptosomes, and these insoluble complexes are enriched in palmitoylated proteins [180]. In agreement with that, Arc was found to undergo palmitoylation at the 94CLCRC98 sequence. Arc induction upon neural activity requires the involvement of transcription factor myocyte enhancer factor 2 (MEF2). In slice cultures of the hippocampus, acute expression of constitutively active transcription factor MEF2 achieved via biolistic transfection induces Arc expression and results in a negative modulatory effect on AMPAR-mediated synaptic transmission manifested by a decreased frequency of mEPSCs. In this model, MEF2-induced depression of mEPSC frequency observed in the presence of palmitoylated Arc was not observed in neurons co-transfected with the non-palmitoylatable Arc mutant protein. Thus, Arc palmitoylation may be essential in some forms of the plasticity at excitatory synapses [180]. Interestingly, only a small portion of total protein may undergo palmitoylation (5–8% and <1% were estimated for Arc and PICK1, respectively) [148,180]. However, as highlighted above, the role of the palmitoylated form of the protein may have a crucial impact on synaptic plasticity.

Cyclin Y in neurons is located mainly within dendritic spines, where it inhibits plasticity-induced AMPA receptor delivery to synapses and LTP in the hippocampus [181]. It has an adverse effect on AMPAR delivery to the postsynaptic membrane in a response to the stimulation with glycine. Moreover, it binds to actin, and by modulating its dynamics, prevents LTP-related structural changes in dendritic spines. The exact subcellular localization is dependent on its lipid posttranslational modification, and therefore, so is the function of the protein. Mutation of cysteine 7 and 8 or treatment with 2-BP prevents cyclin Y from reaching its localization at the plasma membrane near the PSD, resulting in intracellular accumulation of the protein in trans-Golgi network [182]. Much lower density of dendritic spines observed in neurons overexpressing a palmitoylation-deficient mutant of cyclin Y suggests that palmitoylation of cyclin Y plays a regulatory role in spine formation. Overexpression of WT cyclin Y in hippocampal neurons leads to decreased amplitudes of mEPSC and reduced presence of GluA1 on the surface of the spine, and overexpression of palmitoylation-deficient (or myristoylation-deficient) mutants has exactly the opposite effect. Spine shape and the synaptic expression of PSD-95 is also differentially modulated by cyclin Y. In particular, overexpression of wild-type cyclin Y negatively affects those parameters, and expression of the mutant leads to a drastic enhancement in the size of the spine and the presence of PSD-95 at the synaptic membrane. Similarly, contrasting effects of overexpression of cyclin Y WT and palmitoylation-deficient mutant on synaptic properties are also present upon LTP-inducing glycine stimulation. Under such stimulation, neurons overexpressing WT cyclin Y showed decreased exocytosis of GluA1 and clustering of PSD-95 compared to the control, and opposite effects were observed in neurons overexpressing a palmitoylation-deficient mutant of cyclin Y. Interestingly, in glycine-stimulated neurons, overexpression of the WT cyclin Y did not affect spine density, but overexpression of the palmitoylation-deficient (and myristoylation-deficient) mutant protein caused an increase in spine density—an opposite effect to that observed in steady state neurons. In line with the in vitro results, LTP at the Schaffer collateral-CA1 synapses is negatively affected by overexpression of WT cyclin Y in CA1 neurons and further enhanced by expression of the palmitoylation-deficient mutant [182]. Altogether, palmitoylation (and myristoylation) is necessary for cyclin Y to exert its inhibitory role on hippocampal LTP.

Synapsin1 (Syn1) is one of the most abundant presynaptic proteins. It binds to the synaptic vesicles (SVs) and F-actin. ZDHHC5 has been implicated recently in that process [183]. Synapsin-1 may regulate some aspects of the process of SVs recycling, namely, clustering and release of the vesicles. Upon high Ca^2+^ levels during an action potential, Syn1 dissociates from SVs, which leads to a subsequent release of the vesicles into the synaptic cleft. Synapsin1 can also be modified by palmitoylation at multiple cysteines: C223, C360, and C370. Mutation in all three cysteines prevented proper clustering of SVs and reduced the capacity of SV recycling in hippocampal cultured neurons. It was also shown that Syn1 palmitoylation regulated SVs dynamics by facilitating a direct interaction of Syn1 with F-actin (but not SVs), and the release of SVs was accompanied by the depalmitoylation of Syn1. The final piece of evidence supporting the importance of Syn1 palmitoylation in SV-related presynaptic events comes from the ZDHHC5 knockout mice. In this model, decreased palmitoylation of Syn1 was observed, and electron microscopy revealed defects in the SVs’ clustering manifested by decreased density and number of SVs in the presynaptic compartment [183].

Exocytosis of synaptic vesicles is mediated by synaptic SNAP receptor proteins (SNARE). SNARE proteins are localized to vesicles and target membranes, where they bridge and ultimately fuse the membranes. Presynaptic vesicular release largely relies on the SNARE complex, formed by: synaptobrevin-2 (VAMP2), SNAP-25, and syntaxin- 1A or syntaxin-1B (STX1), assisted by modulatory synaptic proteins [184]. Syntaxin1 is an integral plasma membrane protein, and its C-terminal transmembrane domain (TMD) and SNARE motif are separated by a short polybasic juxtamembrane domain (JMD). Syntaxin 1 is palmitoylated in the TMD region [43]. Study on electrophysiological properties of hippocampal neurons expressing palmitoylation-deficient syntaxin-1A (STX1A) single or double cysteine mutants has shown unaltered EPSCs or a readily releasable pool of the neurotransmitter [185]. However, the probability of the release of neurotransmitter was significantly lower, and thus, the frequency of mEPSCs was too. Most strikingly, the double mutant exhibited no spontaneous neurotransmitter release. Altogether, palmitoylation of STX1A’s TMD requires JMD’s cationic amino acids, and loss of palmitoylation impaired spontaneous neurotransmitter release while leaving the Ca^2+^-evoked release intact. It has been proposed that the palmitoylation of two cysteines in the TMD of Syntaxin1 lowers the energy barrier required for the membrane merging of the SNARE complex and spontaneous release of the neurotransmitter [185]. Since the STX1A non-palmitoylatable mutants exhibited a lack of synaptic fatigue upon prolonged synapse stimulation, it appears that palmitoylation of STX1A may play a permissive role in the regulation of short-term plasticity in excitatory synapses.

GABA_A_Rs are the principal mediators of neural inhibition by the neurotransmitter GABA. As we highlight below, it can be postulated that the palmitoylation of GABA_A_R subunits may regulate receptor clustering in the plasma membrane and the efficacy of GABAergic synaptic transmission. Glycine receptors and GABA_A_Rs are clustered in the inhibitory synapses by gephyrin scaffold protein [186]. Dynamic regulation of the number of GABA_A_Rs at synapses provides a key mechanism for the functional plasticity of inhibitory synapses. Palmitoylation of gephyrin at C212 and C284 was shown to be crucial for the clustering of GABAergic receptors at the inhibitory synapses. In addition, ZDHHC12, located in Golgi and primary dendrites, was shown to be the main palmitoylating enzyme of gephyrin [37]. Most importantly, gephyrin palmitoylation is regulated by the neuronal and GABA_A_R activity. Application of the GABA_A_Rs antagonist for 1 h, which is expected to upregulate neuronal activity, decreased gephyrin palmitoylation in vitro and its association with membranes in mouse brain synaptosomes. In contrast, the application of the GABA_A_Rs agonist GABA, which should silence neuronal activity, had the opposite effect. Neurons co-expressing gephyrin-GFP with a palmitoylation-deficient gephyrin construct exhibited reduced miniature inhibitory postsynaptic currents (mIPSCs) amplitudes and prolonged decay, and AMPARs-mediated mEPSCs were not affected [37]. In addition to ZDHHC12, it could be possible that other members of the family may also be involved in the regulation of GABAergic transmission efficacy. Moreover, the impact of palmitoylation on gephyrin may be different in males and females [187]. For instance, ZDHHC7 knockout female mice exhibited increased inhibitory transmission while male ZDHHC7 knockout displayed reduced inhibitory transmission [37,188]. Thus, ZDHHC-mediated palmitoylation of gephyrin may play an important role in regulating the efficacy of GABAergic synapses.

Gi/o-gated G-protein-regulated inwardly rectifying K+ (GIRK) channels also provide a mechanism for inhibitory modulation of synaptic transmission. G-protein-coupled receptor signaling in neuronal cells may be linked with palmitate turnover. Notably, pharmacological inhibition of depalmitoylation caused regulator of G protein signaling (RGS) 7-binding protein (R7BP) to redistribute from the plasma membrane to endomembrane compartments, affecting the formation of RGS complexes with GIRK channels and eventually delayed GIRK channel closure. Thus, palmitoylation/depalmitoylation balance could regulate offset decay times of GIRK channel-mediated currents and thus affect the level of membrane hyperpolarization [189].

#### 2.3.3. Genetically Modified ZDHHCs and Synaptic Function

A great improvement in the understanding of the role of protein palmitoylation in neuronal physiology was gained in studies of transgenic animals. In mice, shRNA-mediated knockdown of ZDHHC2 in the hippocampus negatively affected the interaction of AKAP79/150 and PKA [150]. Additionally, it led to a significant increase in surface localization of GluA1 in basal conditions, but it prevented an enhanced surface in the presence of the protein, normally caused by HFS. The decrease in GluA1’s availability on the spine surface that results from ZDHHC2 knockdown is reflected in impaired LTP. Taken together, these findings implicate the ZDHHC2/AKAP150/PKA signaling pathway in the membrane surface stability of AMPARs and hippocampal LTP [150].

More than 40 different ion channel subunits have been shown experimentally to be S-acylated [190]. However, evidence that protein palmitoylation regulates the gating of ionotropic receptors is scarce. Golgi-specific ZDHHC3 was shown to palmitoylate γ2 subunit of GABA_A_R and mediate the accumulation of GABA_A_Rs at the inhibitory synapses [153,191]. The C157S mutation in ZDHHC3 results in an enzymatically inactive mutant which fails to interact with and participate in trafficking of γ subunits to the membrane and normal accumulation of GABA_A_ receptors at synapses. Consequently, the ZDHHC3 C157S mutant expression resulted in impaired synaptic but also whole-cell GABAergic transmission, but excitatory transmission was unaffected [153]. Similar deficits in GABA_A_R postsynaptic clustering and reduced mIPSCs were observed in co-cultures of wild-type and ZDHHC3 KO cortical neurons [191]. In this model, reduced surface availability of γ2-containing GABARs was observed, and excitatory synapses were unaffected [191]. In summary, ZDHHC3-mediated palmitoylation may regulate the function of synaptic γ2-containing GABARs.

Calcium signaling exerts rapid effects on the phosphorylation and dephosphorylation of ion channels and affects neuronal excitability and synaptic plasticity. ZDHHC3 expressed in oocytes was shown to transport Ca^2+^ through cell membranes, and this transport is not coupled to other ions or metabolic energy [192]. In addition, ZDHHC3-mediated Ca^2+^ transport was voltage-dependent, saturable, pH-sensitive, and inhibited by several alkali earth metals. Similarly, ZDHHC17 expressed in a heterologous system was shown to be a Mg^2+^ ion transporter [192]. Ca^2+^/Mg^2+^ transport mediated by ZDHHC3 and ZDHHC17 is inhibited with 2-BP [158,192,193]. It is not known whether this mechanism could also operate in neuronal cells, where both ZDHHCs are expressed primarily in Golgi and post-Golgi vesicles, but it broadens the potential of these ZDHHCs to affect neuronal plasticity via Ca^2+^/Mg^2+^ availability. Additionally, loss of ZDHHC5 in mice was found to decrease the number of active spines exhibiting Ca^2+^ transients, and the amplitude or the number of Ca^2+^ events per spine remained the same. Loss of ZDHHC5 also increased the AMPAR/NMDAR ratio and affected the number of excitatory synapses [194]. Thus, the transient availability of ZDHHC subtypes could have an impact on the intracellular Ca^2+^ and thus neuronal plasticity.

ZDHHC7 palmitoylates various synaptic and extrasynaptic proteins, such as neural cell adhesion molecule (NCAM) and GABA_A_R subunits. In rats, the expression of ZDHHC7 gene is sex-dependent [195], and this may have implications for brain structure, function, and behavior [188]. Indeed, ZDHHC7 knockout mice were reported to exhibit sex-dependent differences in basal inhibitory and excitatory synaptic transmission. Hohoff and coworkers showed that ZDHHC7 knockout male mice showed less frequent spontaneous EPSCs and prolonged mEPSCs decays in the medial prefrontal cortex [188], and both sexes suffered from significantly less frequent mEPSCs events. Moreover, ZDHHC7 knockout male mice exhibited less frequent IPSCs and mIPSCs, indicating a less efficient inhibitory drive, whereas females had mEPSCs and IPSCs with larger amplitudes. Thus, male ZDHHC7 knockout mice displayed impaired excitatory function and reduced inhibitory transmission, unlike females [188]. In the hippocampus, ZDHHC7 knockout animals did not differ with regard to basal excitatory signals in Sch-CA1 projection or paired-pulse facilitation ratio, indicating no basal difference in synaptic function. However, LTP was impaired or completely abolished in females and males, respectively. Collectively, deficiency of ZDHHC7 impaired LTP in the hippocampi of both female and male animals from the early minutes post LTP induction [188]. In agreement with the above-mentioned results supporting the sex-dependent action of ZDHHC7, we found that several groups of proteins in females which are specifically regulated by ZDHHC7 showed significantly higher enrichment in processes such as synapse structure maintenance (i.e., gephyrin, Rab3a, and Syngap), synaptic vesicle cycle and transport, cellular respiration, and learning [187]. This emphasizes the important role of ZDHHC7 in hippocampal plasticity. Since the hippocampus and medial prefrontal cortex are important parts of the cortico-limbic system, ZDHHC7 could play a role in anxiety, stress, and stress-induced mental disorders. Indeed, ZDHHC7 was shown to play a modulatory role in the brain that leads to sex-specific stress responses, possibly due to estrogen receptor-mediated signaling pathways [196].

Children with microdeletions of the 22q11.2 locus show a high incidence of emotional problems and a spectrum of cognitive deficits and may develop schizophrenia or schizoaffective disorder in adolescence or early adulthood. Chromosomal deficiency Df(16)A containing *ZDHHC8* was engineered in mice to model deletion of part of 22q11.2 locus in the human genome [197]. It has been shown that primary hippocampal neurons from Df(16)A+/− mice have a lower density of dendritic spines and glutamatergic synapses and impaired dendritic growth. In accordance with that, Df(16)A+/− mice exhibited reduced mEPSCs frequency. Similar observations were performed in ZDHHC8 knockout neurons, indicating that this ZDHHC subtype is responsible for the phenotype. ZDHHC8 knockout also exhibited reduced fEPSPs in response to the activation of cortical-channel rhodopsin-expressing afferent fibers [198]. Df(16)A+/− phenotype was restored by enzymatically active ZDHHC8 protein [197]. The underlying mechanism remains unknown but could involve PSD-95, a substrate of ZDHHC8, and/or modulation of RAC-beta serine/threonine-protein kinase/glycogen synthase kinase-3 beta (Akt/Gsk3β) signaling via Cdc42-palmitoylation [198].

In rat primary neuronal cultures, ZDHHC9 is expressed exclusively within somatic Golgi and Golgi satellites associated with both inhibitory and excitatory synapses [36]. ZDHHC9 knockdown resulted in shorter and less complex dendritic arbors and altered the ratio of excitatory to inhibitory synapse inputs to ZDHHC9 mutant cells by reducing gephyrin and the γ2 subunit of GABA_A_R’s cluster density [36]. Decreased palmitoylation of Ras and TC10 small GTPase were implicated in that process. Functional analysis revealed that these changes in neurons were followed by increased amplitude of both mEPSCs and IPSCs, and the frequencies of the events were unaltered. Similarly, ZDHHC9 knockout mice exhibited increased amplitudes of EPSCs and IPSCs; their frequencies were reduced. While it is difficult to unequivocally interpret morphological data in the context of electrophysiological observations in vitro and in slice recordings, ZDHHC9 seems to be essential for supporting network formation and maintaining the excitatory to inhibitory synaptic balance. Indeed, ZDHHC9 knockout mice exhibited more and longer seizure-like events, as visualized with electroencephalographic recordings [36].

ZDHHC17 is enriched in the brain and has been shown to palmitoylate several synaptic proteins i.e., PSD-95, gluA1/2, Glutamate decarboxylase 2 (GAD-65), SNAP-25, and synaptotagmin1, suggesting a role in both pre-and postsynaptic function [199]. In experimental conditions, loss of ZDHHC17 in Drosophila motor neurons led to impairment of neurotransmitter release [38]. The implications of ZDHHC17 knockout have been also studied in mice’s striatal spiny projection neurons (SPNs) and hippocampi [200]. Firstly, the loss of ZDHHC17 resulted in significantly fewer action potentials fired at rheobase (the minimal current amplitude of infinite duration that results in the depolarization threshold). Secondly, while enhanced synaptic facilitation was observed, the number of spontaneous EPSCs and the amplitudes of evoked AMPAR and NMDAR-mediated currents decreased. On the contrary, in another study in post-development, ZDHHC17-deficient mice, synaptic properties of medium-sized spiny neurons (MSNs) in the striatum were examined [201]. A significant decrease in the frequency and a significant increase in the amplitude of sEPSCs were noted in ZDHHC17 knockout neurons, suggesting a reduction in the number of synapses and reduced release probability. Indeed, knockout mice exhibited increased paired-pulse ratio indicative of a lower basal probability of release [201]. In comparison to the striatum, ZDHHC17 loss in the hippocampus resulted in decreased frequency of EPSCs; however, all other tested variables were unchanged. Finally, in hippocampal slices, reduced spine density and impaired LTP were observed in ZDHHC17 knockouts [200]. Altogether, the impact of ZDHHC17 loss in striatal neurons is still debated, and in the hippocampus, it had a negative impact on LTP. These experiments also demonstrate that the loss of a single ZDHHC does not affect all networks equally. Although the exact molecular mechanism remains unknown, the changes observed in ZDHHC17 knockouts may partly be due to reduced palmitoylation of PSD-95 and SNAP25 [200,202].

The impact of ZDHHC17 activity loss was recently studied in zebrafish [203]. Habituation is a type of non-associative learning, considered one of the simplest forms of this phenomenon. It is characterized by a behavioral response decrement as a consequence of repeated stimulation. Habituation is thought to enable animals to engage in more complex forms of learning, as it allows animals to suppress irrelevant stimuli and focus on what is pertinent [204]. Larval zebrafish exposed to repeated high-intensity acoustic stimuli quickly learn to ignore the initially startling stimuli, demonstrating their ability to habituate their behavioral responses. On the molecular level in zebrafish, this process is mediated by reduced Ca^2+^ signaling in a Mauthner neuron, a key component of the startle circuit. Fish with mutated non-functional ZDHHC17 protein display an inability to habituate, which is accompanied by unaffected, robust Ca^2+^ signaling in Mauthner neurons [203]. The observed habituation deficit upon loss of ZDHHC17 catalytic activity is, at least partially, a result of perturbed palmitoylation of a Kv1.1 subunit of Shaker-like voltage-gated K^+^ channel, an important neuronal activity regulator involved in modulating neuronal excitability. In zebrafish, Kv1.1 is mainly palmitoylated at cysteine 238 (which corresponds to cysteine 243 in humans), and mutation of that cysteine almost entirely diminishes Kv 1.1 palmitoylation. Indirect evidence points to the palmitoylation of Kv 1.1 acting as a habituation promoter by regulating the protein’s localization. In ZDHHC17-mutant zebrafish, Kv 1.1 displayed a significantly reduced presence in its target location—the presynaptic terminals of the so-called axon cap in the Mauthner axon. What is worth mentioning, even though the experiments were performed in the larval form of zebrafish, is that the habituation deficit in ZDHHC17 mutant fish was independent of developmental processes but rather stemmed from a perturbed acute response to changes in neural activity [203].

#### 2.3.4. Depalmitoylation and Synaptic Function—Lessons from PPT1 Knockout Mice

Mutation of the depalmitoylating enzyme palmitoyl-protein thioesterase 1 (PPT1) causes infantile neuronal ceroid lipofuscinosis (CLN1), a pediatric neurodegenerative disease. Therefore, recently the role of PPT1 was studied in synapse development in the mouse visual cortex [205]. GluN2B-containing NMDARs are expressed neonatally and display prolonged decay kinetics, which allows comparatively increased Ca^2+^ influx and thus facilitates forms of synaptic plasticity. With development, a switch from GluN2B towards GluN2A-containing NMDARs is observed [206]. It was shown that this developmental NMDAR subunit switch is stagnated in the PPT1-/- mouse’s visual cortex [205]. In addition, PPT-/- neurons exhibited prolonged decay kinetics of evoked NMDAR currents and dendritic spine morphology in vivo. Lack of PPT1 also resulted in prolonged and diffused Ca^2+^ influxes; increases in calcium transients, frequency, and diffusion distance predominantly in the extrasynaptic region; and enhanced vulnerability to NMDA-induced excitotoxicity, reflecting the predominance of GluN2B-containing receptors. These functional changes were associated with hyper-palmitoylation of GluN2B and Fyn kinase, which regulates surface retention of GluN2B in PPT1 knockout neurons. Altogether PPT1 seems to play a critical role in postsynaptic maturation by facilitating the GluN2 subunit switch.

Hippocampal neurons with PPT1 loss displayed structural and functional deficits, which included decreased dendritic tree complexity and lower dendritic spine density, and functional changes, such as decreased frequency of miniature excitatory synaptic currents. However, PPT1 knockout did not affect the amplitude and kinetics of excitatory synaptic currents [207]. In agreement with that, Koster and coworkers also observed a reduced frequency of mEPSCs in PPT1 knockout neurons from the 2/3 layer of the developing visual cortex [208]. This is in agreement with the idea that PPT1 is crucial for the maintenance or development of synaptic connections. By affecting Ca^2+^ transients through regulating GluN2B palmitoylation status, PPT1 activity in neurons may significantly affect various forms of synaptic plasticity [205]. Moreover, PPT1 knockout mice exhibited significantly impaired HFS-LTP. In addition, the magnitude of synaptic scaling following chronic deprivation of neuronal activity with TTX was significantly exaggerated in the primary cortical neurons of PPT1 knockout mice due to hyperpalmitoylation of GluA1 and membrane incorporation of calcium-permeable AMPARs [208]. Altogether, PPT1 may be crucial in various forms of synaptic plasticity, following enhanced neuronal activity, and we are just beginning to uncover a plethora of PPT1 functions.

PPT1 has also recently been found to be involved in the palmitoylation-dependent regulation of GABA_A_R function [209]. In PPT1-deficient mice, the GABA_A_R α1 subunit was the primary substrate for PPT1 in the synapses, unlike γ2 or β subunits or gephyrin, a scaffold protein shown to undergo palmitoylation. Lack of PPT1 led to hyper-palmitoylation and increased the membrane availability of GABA_A_Rs in the hippocampus. Functionally, this resulted in increased amplitudes of both evoked inhibitory currents’ responses and mIPSC and increased mIPSC frequency, while the excitatory synaptic transmission was unaffected. Importantly, the application of depalmitoylation agent NtBuHA reversed those effects [209]. Overexpression of PPT1 in human neuronal-like cells resulted in the reduced transcription of voltage-gated calcium and potassium channel subunits and reduced Ca^2+^ influx [210]. This indicates a potential role of PPT1 in modulating neuronal excitability. A recent large proteomic study identified PPT1 substrates at the synapse, which included channels and transporters, G-protein–associated molecules, endo/exocytic components, synaptic adhesion molecules, and mitochondrial proteins [70].

Neurotransmission is regulated by repeated cycles of exocytosis and endocytosis of the synaptic vesicles and palmitoylation of presynaptic release machinery proteins, i.e., SNAP25, vesicle-associated membrane protein 2 (VAMP2), and syntaxin 1, which may play critical roles in neurotransmitter release and the maintenance of synaptic strength [58]. However, the palmitoylated proteins associated with the synaptic vesicles must undergo depalmitoylation to be detached from the membrane, recycled, and/or degraded. It has been shown that PPT1 is present in synaptosomes and synaptic vesicles. Moreover, in PPT1 knockout mice, significantly lower levels of soluble VAMP2, SNAP25, and syntaxin1 were detected, while membrane-associated levels were increased, leading to persistent membrane association of synaptic vesicle proteins [211]. A brief 30–60 s stimulation of neurons with a high K^+^ solution and observation of the de-staining of presynaptic release marker FM1-43FX for the next 7 min led to the conclusion that PPT1 knockout caused a decline in the number of readily releasable synaptic vesicles [211]. Altogether, a picture emerges that PPT1-mediated depalmitoylation controls persistent membrane anchorage of the palmitoylated synaptic vesicles proteins and regulates the recycling of the vesicle components that normally fuse with the presynaptic plasma membrane during exocytosis [211].

## 3. Palmitoylation in Learning and Memory

The ability of neurons, especially in assemblies, to respond to external stimuli adaptively, is well established in vivo, as it forms the basis for all animal learning [212]. The formation of memory involves encoding, storing, consolidating, and retrieving information; and this is largely facilitated by neuronal plasticity [213]. Considering the supportive role of protein palmitoylation in synaptic plasticity, it is expected that it also plays an important role in learning and memory formation. There is growing evidence that learning and memory require palmitoylation and are associated with an increased abundance of palmitate in neurons. On the other hand, excess palmitate from diet or exogenous application has a negative effect on cognitive functions. Below, we discuss studies supporting this relationship.

While an excess of exogenous palmitate seems to be detrimental to cognitive performance, endogenous protein palmitoylation supports synaptic plasticity, learning, and memory in young brains. The first line of evidence comes from studies on transgenic animals. ZDHHC5 is located primarily in excitatory synapses. ZDHHC5 gene-trap mice with low expression levels of this enzyme exhibited a marked deficit in contextual but not cued fear conditioning, an indicator of defective hippocampal-dependent learning [214]. Another member of the ZDHHC family, ZDHHC2, was recently reported to alter the cognitive capabilities of mice [150]. In the hippocampi of WT mice after fear conditioning, there is an increased level of palmitoylation of both ZDHHC2 and AKAP79/150. The ZDHHC2 shRNA-mediated knockdown reversed AKAP79/150 palmitoylation and led to impaired fear-memory formation. These results imply a supportive role of ZDHHC2 in learning and memory [150]. Constitutive loss of ZDHHC17 was shown to significantly impair novel-object location memory (spatial memory), and object recognition memory stayed intact [200]. Similarly, ZDHHC9 knockout mice exhibited mild impairment in spatial learning [215]. Altogether, evidence from transgenic models indicates that ZDHHC2, ZDHHC5, and ZDHHC17 may be involved in hippocampal learning. In addition to that, most recently, Nasseri and coworkers used the comparative mass spectrometry approach to analyze mouse brain palmitoylome showing broad and significant changes in the brains following learning in fear conditioning paradigm [41].

Spatial learning and memory deficits were observed in PPT1 loss-of-function mutation knock-in mice. The underlying mechanism involved hyperpalmitoylation of GABA_A_Rs and disruption of GABAergic neurotransmission (see Section 2.3.4). The effect of PPT1 dysfunction and the enhanced palmitoylation were reversed by the application of depalmitoylating agent NtBuHA, further supporting the notion that palmitoylation remains a crucial factor in hippocampal spatial learning [209]. In addition, a balance in palmitoylation/depalmitoylation cycles was shown to play an important role in the fear-conditioning-induced strengthening of synaptic transmission. In this behavioral paradigm, elevation in hippocampal global palmitoylation was observed. Treatment with the palmitoylation inhibitor 2-BP reduced fear-memory acquisition in rats, and an inhibitor of the depalmitoylating enzyme APT1 showed a tendency to facilitate fear-memory formation in vivo, although this effect was not statistically significant [161].

There is also a relationship among local palmitate abundance and neuronal activity, learning, and memory. In the cultured rat cortical neurons, palmitic, stearic, and myristic acid constituted 86% of all 19 fatty acids, and their concentrations ranged from 1 to 3 µM. Importantly, most studied free fatty acids (FFAs) were enhanced 15 min after membrane depolarization with a high K^+^ concentration, suggesting that neuronal activity may be mirrored by the varying availability of palmitic acid in the cell [12]. In vivo, a targeted lipidomics approach aimed to characterize FFAs, and phospholipids in the rat brain revealed that saturated FFAs (6:0–24:0) constitute the majority of the FFAs across all brain regions. Palmitic acid was mostly enriched in brain areas associated with fear learning and memory, such as the amygdala, and to a much lesser extent in the hippocampus, and the minimum amounts were detected in the cerebellum [216]. Moreover, auditory fear conditioning led to an increase in saturated FFAs (including palmitic acid) in the amygdala and prefrontal cortex 2 h post-training. Moreover, this change was not observed upon application of NMDAR blocker CPP, indicating a further link between mechanisms of neuronal plasticity and palmitoylation in learning [216]. Importantly, it appears that this learning paradigm upregulates not only palmitic acid, but all studied FFAs with 6–24 carbon atoms, and in particular, myristic acid (14:0) [216].

Brain palmitic acid is sourced near equally from both the dietary uptake and local brain de novo lipid synthesis [217,218]. Exogenous palmitic acid has toxic effects on brain cells, and works in that field have been recently extensively reviewed [219]. Obesity and a chronic high-fat diet rich in palmitic acid have been associated with reduced cognitive function in animals and humans [216]. Moreover, palmitate is increased in the cerebrospinal fluid (CSF) of overweight and obese patients with amnestic mild cognitive impairment [220]. Thus, the cerebrospinal fluid palmitate level inversely correlates with cognitive performance. In agreement with that, it was shown that infusion of palmitate into the ventricles of mice resulted in reduced field-excitatory postsynaptic potentials in the CA1 hippocampal region, a typical sign of reduced excitatory synaptic transmission. In addition, high palmitate also reduced HFS-LTP recorded in vivo [220]. In addition, the same animals subjected to novel object recognition and novel object learning tests presented memory deficits after palmitate infusion [220]. These effects of palmitate were observed together with increased hippocampal inflammation, astroglial and microglial activation, and tumor necrosis factor α (TNF-alfa) signaling [220]. Interestingly, when oleate was used instead of palmitate, no effect on learning or LTP was observed, indicating specific actions of palmitate. Altogether, the negative effects of excess palmitate contributing to decreased cognitive performance may be linked to brain inflammation and impaired insulin signaling in overweight humans.

Mutation of cysteines in a protein that is a substrate to palmitoylation is another strategy to investigate the role of this process in learning and memory. Mice with GluA1C811S mutant protein deficient in palmitoylation exhibited enhanced seizure susceptibility and LTP-induced spine enlargement [164]. Furthermore, GluA1C811S mice were shown to have no alterations in locomotion, sociability, or depression-related behaviors, or spatial learning and memory. However, they exhibited prolonged contextual fear memory. Thus, palmitoylation of the AMPA receptor’s GluA1 subunit at C811 may regulate the formation and extinction of contextual fear memory, unlike cued or spatial memory [221].

There is a line of evidence suggesting that the palmitoylation of certain proteins may be linked to learning and memory. For instance, δ-catenin is highly expressed in the brain and is particularly abundant in dendritic spines. It is responsible for linking cadherin to the actin cytoskeleton or postsynaptic scaffold molecules, including PSD-95. Neuronal activity promotes the recruitment of δ-catenin to synapses. When already in the synapses, palmitoylated δ-catenin is required for cLTP-induced AMPA receptor synaptic insertion and upregulation of synaptic transmission [30]. Interestingly, in vivo, palmitoylation of δ-catenin was significantly increased 1 h after hippocampal-dependent, contextual fear conditioning. Although δ-catenin-palmitoylation-deficient animals were not tested for cognitive impairments, it is possible that in vivo synapse plasticity associated with learning events requires palmitoylation of that protein [30]. Similarly, it was proposed based on in vitro studies that stability and subcellular localization of SynDIG1, and indirectly, AMPAR, may be regulated by activity-dependent palmitoylation [157]. Indeed, palmitoylation of proteins of this family may support learning and memory, since palmitoylated Syndig4 proline-rich transmembrane protein 1 (PPRT1) was found recently to be enriched following learning [41].

As mentioned previously, the palmitoylation of cyclin Y plays a crucial role in the regulation of the magnitude of synaptic plasticity in the hippocampal excitatory synapses [182]. In animals that underwent Morris maze spatial learning, both expression and palmitoylation levels of cyclin Y were reduced in the hippocampus and cortex, suggesting that cyclin Y is actively regulated during or immediately after completion of spatial learning. Interestingly, overexpression of cyclin Y resulted in decreased performance in the Morris water maze test. This effect was not observed when a non-palmitoylatable mutant of cyclin Y was expressed. Mice overexpressing cyclin Y mutant C7,8S also showed a tendency toward a better response regarding latency to the platform zone compared to mice overexpressing WT protein. Interestingly, the effect of palmitoylation and myristoylation on learning was saturation on day 5 of training, which suggests that these post-translational modifications could only affect the speed but not the capacity of learning [182]. Altogether, palmitoylation and myristoylation of cyclin Y regulate synaptic and cognitive functions.

## 4. Palmitoylation in Aging and Diseases Associated with Cognitive Decline

Numerous studies link defects in protein palmitoylation and the aberrant activity of palmitoylating/depalmitoylating enzymes with human neurological disorders and neurodegenerative diseases. Several reviews provide great references on that matter [9,222,223,224]. Below we discuss to what extent protein palmitoylation may contribute to pathomechanism of selected brain diseases associated with impaired neuronal plasticity and cognitive functions. Moreover, the pathomechanism of several human diseases modeled in mammals whose symptoms entail cognitive impairment involves elements of the palmitoylation machinery. Some other diseases not described in this review, such as Parkinson’s disease and epilepsy, have been also linked to altered palmitoylation status of neuronal proteins, and in particular, ZDHHC8 [222,225,226,227]. Below, we also discuss palmitoylation in the context of aging.

### 4.1. Palmitoylation and Aging

Cognitive abilities decline with age in mice and humans; however, the underlying mechanism is complex and debated. Several brain regions coordinate memory acquisition, consolidation, and retrieval. The prefrontal cortex is primarily involved in executive functions, including short-term or working memory and cognitive flexibility (reviewed in [228]). In the mouse brain, total protein expression of NMDAR subunits was previously reported to decrease with age (reviewed in [229]), in contrast to palmitoylation levels of these receptors. In a study by Zamzow and coworkers, young and old mice were behaviorally characterized with the Morris water maze test, and following the test, their brain samples were analyzed for palmitoylation. It was found that aging did not alter the fatty acid transport proteins or the availability of palmitate or palmitoyl-CoA; however, there were increases in the levels of palmitoylation of GluN2B, GluN2A, and Fyn in the prefrontal cortex, which most likely resulted from a disturbed cellular palmitoylation cycle and had detrimental effects on reference memory. An increase in palmitoylation was associated with reduced performance in reference memory tests and executive functioning but did not account for cognitive differences between young and old mice. Interestingly, aging led to an increase in palmitoylation of APT1 in the prefrontal cortex, which could trap the protein on the Golgi apparatus and this way contribute to the observed deficits in reference memory [230].

### 4.2. Neurodegenerative Diseases

#### 4.2.1. Neuronal Ceroid Lipofuscinoses

The neuronal ceroid lipofuscinoses (NCLs) (also known as Batten disease) are a class of inherited nervous system disorders associated with mutations in 13 genes (CLN1 to CLN14) that most often begin in childhood and interfere with a cell’s ability to recycle a cellular residue called lipofuscin [231]. NCL-associated proteins are localized to lysosomes, endoplasmic reticulum, and other cellular locations; but CLN gene mutations lead primarily to dysfunction in lysosomes. One class of diseases called infantile neuronal ceroid lipofuscinosis (CLN1 disease) is caused by a mutation in *CLN1*, which encodes palmitoyl-protein thioesterase 1 (PPT1) [223] and may have several phenotypes and types of symptom progressions [232]. In the infantile form, difficulties in acquiring skills such as standing, walking, and talking occur between 6–12 months, and may progress with age into vision loss, seizures, psychomotor deterioration, and premature death [233].

The molecular mechanism explaining how PPT1-deficiency may affect cell physiology was studied in various systems. In Drosophila, loss of PPT1 causes defects in endocytic trafficking [234] and exocytosis and endocytosis of synaptic vesicles [235]. In mice, PPT1 knockout resulted in reduced expression of presynaptic proteins (e.g., SNAP25), a smaller number of synaptic vesicles, and abnormal expression of NMDAR, contributing to excitotoxicity [223]. PPT1 deficiency caused persistent membrane anchorage of the palmitoylated SV proteins, which hindered the recycling of the vesicle components that normally fuse with the presynaptic plasma membrane during SV exocytosis both in postmortem brain tissues from patients and in brain tissues from PPT1 knockout mice [211]. Altogether, loss of PPT1 in infantile ceroid lipofuscinosis may result in degeneration of excitatory and inhibitory cells and impaired synaptic transmission [223,236]. Recent developments in targeted therapies against neuronal ceroid lipofuscinoses have been reviewed [233,237].

#### 4.2.2. Huntington’s Disease

Huntington’s disease (HD) is an adult-onset autosomal-dominant neurodegenerative pathology characterized by progressive cognitive decline and other psychiatric disturbances, and motor dysfunction [238]. Huntingtin (HTT) acts as a scaffold protein and interacts with a great number of proteins (over 350) involved in diverse cellular processes, including transcription, trafficking, cytoskeleton dynamics, and autophagy [239]. Mutation in the gene for HTT leads to an abnormal protein with expanded polyglutamine in its N-terminal domain and impaired protein-protein interactions.

HD is recognized as a disease of altered palmitoylation for several reasons [9,224,240]. Firstly, HTT is palmitoylated at C214 by ZDHHC17 and its paralog, ZDHHC13, also known as huntingtin-interacting protein (HIP14) and HIP14-like, respectively. The palmitoylation of HTT is affected by the HD mutation. In the YAC128 transgenic mouse model expressing human mutant HTT, the mutant protein was shown to be less palmitoylated compared to the WT protein [241]. Secondly, transgenic mice lacking ZDHHC17 or ZDHHC13 mimic the pathophysiology of HD, and the mutant HTT negatively regulates the ZDHHCs, making them less active [202,242,243]. Consequently, the substrates of ZDHHC17 and ZDHHC13, such as postsynaptic density protein PSD-95 and presynaptic protein SNAP25, are less palmitoylated, leading to perturbed synaptic function.

Recently, several articles advanced our understanding of the palmitoylation in the pathogenesis of HD. In YAC128 mice, there’s increased striatal neuronal NMDAR-mediated current and excitotoxicity due to changes in membrane trafficking of the protein including enhanced extrasynaptic GluN2B-type NMDAR (NMDAR2B) surface expression, which is thought to promote a shift towards activation of cell death-signaling pathways underlying HD pathology. NMDAR2B surface expression is modulated by palmitoylation at two cystine clusters in the C-terminal domain of GluN2B. Palmitoylation at the cysteine cluster II is mediated by ZDHHC13 and when reduced leads to an increase in plasma-membrane incorporation of NMDAR2B at the extrasynaptic sites. Taken together, the perturbed function of huntingtin-associated PAT, ZDHHC13, in YAC128 mice, promotes NMDA toxicity in the striatal neurons, thereby increasing their susceptibility to apoptosis and contributing to HD pathology [244].

Mutant HTT-induced neurotoxicity was shown to be significantly reduced by increasing the HTT palmitoylation level by inhibition of depalmitoylating APTs (acyl-protein thioesterases) with palmostatin B in YAC128 neuronal cultures [243]. More precisely, it is the function of APT1 rather than APT2 that seems to be involved in the HD pathology. APT1 activity was found to be elevated in CAG140 mice (an alternative transgenic HD mouse model), and the APT1-selective inhibitor ML348 was shown to rescue trafficking, synapse number, and neuronal function in CAG140 neuronal cultures and ameliorate the behavioral phenotype and neuropathology in CAG140 mice [245]. Importantly, damaged corticostriatal circuitry, a hallmark of HD, is mainly due to impaired transport of the brain-derived neurotrophic factor (BDNF) [246,247] and ML348 restored trafficking of BDNF in CAG140 neurons and cortical neurons derived from iPSCs of a HD patient [245].

Caspase-6 (CASP6), a cysteine protease, plays a key role in axonal degeneration in HD [248]. HTT is a substrate of CASP6, and the cleavage of mutant HTT by CASP6 results in the generation of a toxic N-terminal HTT fragment. Impaired palmitoylation contributes to CASP6-related pathogenesis in HD. CASP6 is palmitoylated at two cysteines by ZDHHC17, and palmitoylation negatively affects CASP6 activity, and therefore, its ability to generate the toxic HTT fragments. CASP6 palmitoylation was found to be decreased in brains of YAC128 mice [249], which can be explained by the diminished activity of ZDHHC17 in HD.

#### 4.2.3. Alzheimer’s Disease

Alzheimer’s disease is a neurodegenerative brain disorder characterized by the presence of abnormal clumps (called amyloid/neuritic plaques) and neurofibrillary tangles. There is no preventive drug or cure currently available (reviewed in [250,251]). Since the damage takes place initially in the parts of the brain which are involved in memory, the symptoms include the decline in cognitive functions such as memory, processing speed, reasoning, and problem-solving; and altered behavior, including paranoia, delusions, and loss of social appropriateness. The proteolytic processing of amyloid precursor β-protein (APP) by a β-Site APP cleaving enzyme-1 (BACE1), a trans-membrane aspartyl protease, results in the formation of β-amyloid peptides (Aβ). These Aβ are known to be axonally transported in neurons and accumulate in dystrophic neurites near cerebral amyloid deposits, leading to the pathogenesis of Alzheimer’s disease.

Several proteins implicated in the development of Alzheimer’s disease are regulated by palmitoylation, and the role of palmitoylation in the pathomechanism of Alzheimer’s diseases has been reported [222]. The APP trafficking is regulated by a protein alcadein containing an Asp–His–His–Cys (DHHC) domain and APP interacting DHHC protein (AID)/DHHC-12. An AID/DHHC-12 mutant of which the enzyme activity was impaired by replacing the DHHC sequence with Ala–Ala–His–Ser (AAHS) showed increased non-amyloidogenic α-cleavage of APP, suggesting that protein palmitoylation was involved in the regulation of non-amyloidogenic α-secretase activity [252]. APP was also reported to undergo palmitoylation itself at two cysteine residues, C186 and C187, and DHHC7 and DHHC21 overexpression increased Aβ production and APP palmitoylation [253,254]. In addition, BACE1 is a substrate for ZDHHC3, ZDHHC4, ZDHHC7, ZDHHC15, and ZDHHC20 and may be palmitoylated at four cysteine residues: C474, C478, C482, and C485 [255]. However, the role of BACE1 palmitoylation in APP processing is debated [222]. Since APP palmitoylation enhances amyloidogenic processing by targeting APP to lipid rafts and enhancing its BACE1-mediated cleavage [253], inhibition of palmitoylation of APP formation by specific palmitoylation inhibitors was proposed as a strategy for the prevention and/or treatment of Alzheimer’s disease [253]. An increase in palmitoylated APP levels was also found to increase APP dimerization in cells, along with an increase in Aβ generation [254]. Overall, studies of APP imply that APP palmitoylation, APP dimerization, and aging contribute to the development of Alzheimer’s disease.

Postsynaptic density protein (PSD-95) levels are known to normally get diminished in aging and neurodegenerative disorders. α/β-Hydrolase-domain-containing protein 17 (ABHD17) in neurons selectively reduces PSD-95 palmitoylation and synaptic clustering of PSD-95. ABHD17 inhibition was shown to increase the level of PSD-95 by selectively blocking its depalmitoylation and reversing the Aβ effects on synapses [141]. Interestingly, increasing PSD-95 by epigenetic editing in aged or Alzheimer’s disease model mice has been shown to enhance cognitive function [256]. Therefore, selectively blocking PSD-95 depalmitoylation may serve as a viable therapeutic option for developing a treatment for Alzheimer’s disease. Most recently, to address this issue, Dore et al. performed experiments in organotypic hippocampal slices with an increased Aβ level. It was found that palmostatin B (an ABHD17 inhibitor) rescued Aβ-induced synaptic depression and Aβ-mediated effects on dendritic spines [257]. It was concluded that increasing PSD-95 synaptic content via palmitoylation may protect synapses from Aβ-induced synaptic deficits by reducing NMDA receptor metabotropic function.

### 4.3. Neuropsychiatric Disorders

#### 4.3.1. Major Depressive Disorder

Clinical depression, or major depressive disorder (MDD), is a common mental disorder described with a spectrum of symptoms, such as pervasive low mood and vegetative symptoms, along with deficits in cognitive and psychomotor processes [258]. The underlying mechanism is extremely heterogeneous and complex, which hinders the development of treatments that are effective for all depressed individuals. Abnormalities in the neurotransmitter serotonin levels and efficacy of serotonergic neurotransmission are thought to play a crucial role in MDD (reviewed in [259]), and thus, the primary treatment involves the use of selective serotonin reuptake inhibitors (SSRIs). The role of serotonin receptors in the pathomechanism of depression has been discussed but remains debated (reviewed by [260]). Existing evidence links palmitoylation of serotonin receptors with MDD (reviewed by [9]). Most recently, it has been shown that serotonin 1A receptors (5-HT1AR) are implicated in MDD. In particular, palmitoylated 5-HT1AR has been shown in human and rodent brains and is a substrate for ZDHHC21. Depressive-like behavior in animals has been linked to reduced brain ZDHHC21 expression and attenuated 5-HT1AR palmitoylation. Furthermore, analysis of post-mortem brain samples of MDD patients who died by suicide were found to corroborate results from animals. This suggests that downregulation of 5-HT1AR palmitoylation is a mechanism involved in depression [261]. Altogether, MDD may be associated with reduced brain ZDHHC21 expression and attenuated 5-HT1AR palmitoylation.

Serotonin transporters are responsible for the clearance of serotonin from the extraneuronal space. Serotonin modulates mood, aggression, motivation, appetite, sleep, cognition, and sexual function. An SSRI, fluoxetine, which has been used to enhance serotonin efficacy, has been recently implicated in affecting cellular glucose uptake by palmitoylation of glucose transporters GLUT3 and GLUT1. This suggests that some actions of fluoxetine may be ascribed to palmitoylation of non-target proteins [262].

Another study introduced the relation of peripheral spared nerve injury (SNI)-induced interleukin 6 (IL-6) overexpression with depression-like behaviors. Astrocytes that serve as an important supporting cell in modulating glutamatergic synaptic transmission were demonstrated to release IL-6 in basolateral amygdala (BLA) as a result of SNI. This enhanced the abundance of ZDHHC2 in the synaptosome and ZDHHC3 in the Golgi apparatus, promoting PSD-95 palmitoylation to thereby increase the recruitment of GluR1 and GluN2B at the synapses. Suppression of IL-6 or PSD-95 palmitoylation attenuated the synaptic accumulation of GluR1 and GluN2B in BLA and improved depression-like behaviors that were induced by SNI [263].

#### 4.3.2. Schizophrenia

Schizophrenia is a chronic mental disorder interfering with the ability to think clearly, management of emotions, and decision making. Various PTMs, particularly palmitoylation, have been found to be differentially expressed in several pathways that are hypothesized to be dysregulated in schizophrenia [222]. For example, membrane association and transport of phosphodiesterase 10A (PDE10A) throughout dendritic processes in primary mouse striatal neuron cultures require palmitoylation at C11 of PDE10A2, likely by the palmitoyl acyltransferases ZDHHC7/19 [264]. Many reports also suggest that the abnormalities of neurotransmitter receptor trafficking, targeting, dendritic localization, recycling, and degradation in the brain during schizophrenia are due to dysregulation of protein palmitoylation. A mass spectrometry study which identified 219 palmitoylated proteins in a normal human frontal cortex showed significant reductions in the levels of vesicular glutamate transporter 1 (VGLUT1), myelin basic protein (MBP), and Ras family proteins when palmitoylation was assayed and compared in the dorsolateral prefrontal cortices from 16 schizophrenia patients and their pair-matched comparison subjects [265]. In addition, rats chronically treated with haloperidol showed the same pattern and unchanged extent of palmitoylation. These two results suggest that there is a 20% reduction in the palmitoylation level of many proteins in the frontal cortex in schizophrenia, which is not due to chronic antipsychotic treatment [265].

Similarly, a cohort study of approximately 36 individuals was performed to substantiate a bivariate correlation analysis of PPT1 enzymatic activity in first-episode psychosis (FEP) patients with a psychiatric assessment score. It established that the higher enzymatic PPT1 activity in FEP schizophrenia patients is associated with increased positive and negative syndrome scaling values, indicating more serious rates of developing psychosis [266]. Interestingly, there are also some genetic deletions disturbing regular palmitoylation processes that are associated with the risk of developing schizophrenia. The deletion of *ZDHHC8*, which codes for PAT, contributed to the development of schizophrenic symptoms, including alterations in cellular, anatomical, and behavioral levels. Basically, a ZDHHC8 knockout mice demonstrated reduced palmitoylation of Cdc42 and Ras-related C3 botulinum toxin substrate 1 (Rac1), altering neuron polarity and showing some deficits in the connectivity between the hippocampus and the medial prefrontal cortex, eventually contributing to impaired working memory performance [198]. To better understand the genetic basis of schizophrenia, a large-scale integrative analysis of genome-wide association study (GWAS) and expression quantitative trait loci (eQTLs) data detected *ZDHHC18* and *ZDHHC5* as two of the five significant genes associated with schizophrenia [267].

### 4.4. Neurological Disorders

#### Chromosome X-Linked Intellectual Disability

Intellectual disability is a neurodevelopmental disorder occurring in 1–3% of the general population [268]. Several clinical signs/symptoms have been associated with X-linked intellectual disability, including developmental delay with severe speech delay, dysmorphic features, epilepsy, or abnormal brain visualized with magnetic resonance imaging. However, the most prevalent features are the deficits in intellectual functions, such as learning and problem solving, and in adaptive functions, such as practical and social skills [269,270]. In humans, X-chromosome-linked intellectual disability has been linked to over 100 gene mutations, including *ZDHHC9* and *ZDHHC15* genes [271,272,273]. Mutations in *ZDHHC9* are more prevalent in males [274] and may impair ZDHHC9 enzyme activity by affecting its auto-S-fatty acylation levels [275]. Interestingly, ZDHHC9 knockout male mice exhibited neurological impairments observed in humans, such as hypotonia and impaired performance in a spatial learning task [215]. In addition, ZDHHC9 knockout mice exhibited reduced corpus callosum volume, similar to that seen in humans with a ZDHHC9 mutation [276]. This proves that ZDHHC9 activity may be crucial for maintaining some of the higher brain functions, including cognitive abilities. At the current state, the molecular substrates for ZDHHC9 in these processes remain unknown. Finally, the ZDHHC9 mutant mouse line could provide a model system to better understand the mechanism of pathological neurodevelopmental changes that lead to intellectual disabilities associated with X-chromosome mutations.

## 5. Conclusions and Final Remarks

This review of the current literature indicated that protein palmitoylation is altered upon changes in neuronal activity and following induction of neuronal plasticity. On the other hand, interference with palmitoylation/depalmitoylation in neurons may modify basal excitatory and inhibitory synaptic transmission and synaptic plasticity. Clearly, protein palmitoylation may regulate the magnitudes of experimentally observed synaptic LTP and LTD. Chemical and electrical stimuli leading to increased palmitoylation of one family of synaptic proteins may result in depalmitoylation of another, but how these processes are directed on the molecular level remains largely unexplored. The majority of studies published so far described glutamate AMPA and NMDA receptors and GABA_A_Rs and their respective scaffolding and auxiliary proteins. However, many other ligand-gated and G-protein-coupled receptors (GPCRs) in the central nervous system undergo regulation by palmitoylation [97,277]. Future research will show the importance of palmitoylation in functional synaptic plasticity involving GPCRs.

Development of new tolls or standardizations of existing tools will be helpful to improve our understanding of the role of palmitoylation in brain function. For instance, there is no consensus on the concentration and time of incubation with palmitoylation inhibitor 2-BP. The doses used ranges from 10 to 50 μM 2-BP for 8 h to overnight treatment [160,278,279,280] to even 250 μM for 1 h in RAW264 cells [281]. It seems important, as there is information about the cytotoxic effects of 2-BP [282] and also the toxicity and teratogenicity of 2-BP for neural development in vitro in the zebrafish model [283]. Employment of genetic engineering has given the opportunity to provide evidence linking palmitoylation machinery with learning and memory.

Several factors efficiently slow down the research on S-palmitoylation in the brain. The substrate specificity of PATs and thioesterases is poorly understood. Publications of brain palmitoylome facilitate the search for protein candidates, but still, every protein requires further in vitro or in vivo assays. On the other hand, some neuronal proteins are palmitoylated by several members of the PAT family. Thus, with such redundance, animal models with constitutive knockout of one gene carry a risk of data misinterpretation. In addition, many proteins are palmitoylated on more than one cysteine. The site of palmitoylation and number of palmitoylated residues may each have a significant impact on protein function. However, most methods used for the detection of palmitoylated proteins do not have sufficient resolution. The use of mass spectrometry and the acyl-PEG method largely facilitates this work. The advances in techniques and improvements in technologies have encouraged neurobiologists to explore, curate, and analyze expression patterns of various palmitoylating and depalmitoylating enzymes in mouse brains from publicly available RNAseq datasets. We provided a comprehensive overview of palmitoylated proteins’ distribution in the mouse’s nervous system [68]. It is equally important to note that ZDHHCs are also substrates for PTMs which change their stability, localization, and function (see [284] for a review). Furthermore, S-palmitoylation may be affected by other PTMs, which can occur on the same protein (i.e., phosphorylation) [285], or even the same cysteine residue (S-nitrosylation) [286]. The interplay between PTMs may be another layer of complexity of protein function control and highlight the need for performing experiments in advanced models which take that into account. Finally, the tools to analyze activity and localization of the members of the palmitoylation machinery are largely lacking—i.e., enzyme-specific antibodies and visualization aids. Enzyme-specific inhibitors that could operate both in vitro and in tissue are largely lacking. Some recent discoveries may significantly change that situation (reviewed in [287]; see also [87]). The recent use of inhibitors of thioesterases in experimental models of Huntington disease and Alzheimer’s disease has been shown to have the potential therapeutic effects [245,257]. Therefore, development of highly specific drugs interfering with palmitoylation/depalmitoylation cycles may be helpful for planning therapies.

Several questions still remain to be addressed in the future research: (i) What are the extracellular chemical cues and intracellular pathways driving protein-specific palmitoylation/depalmitoylation cycles? (ii) What is the time-course of synaptic protein palmitoylation following presentation of the stimulus? Since ZDHHC3 could transport Ca^2+^ in voltage- and pH-dependent manner [192], and ZDHHC17 could be involved in Mg^2+^ transport [193], are there non-canonical functions of ZDHHC enzymes in brain cells?

## Figures and Tables

**Figure 1 cells-12-00387-f001:**
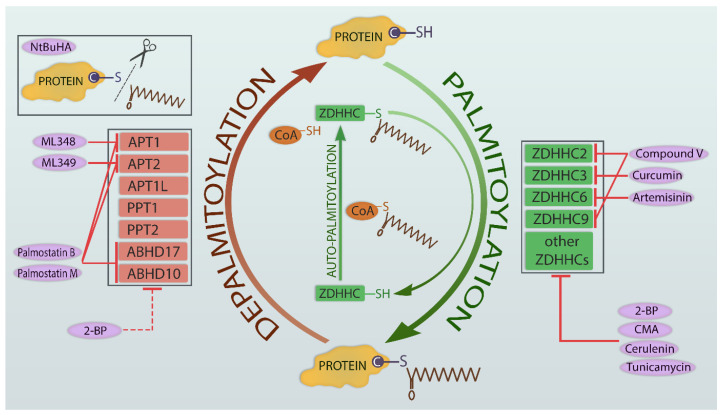
Symbolic illustration of the palmitoylation/depalmitoylation cycle with key enzymes and their inhibitors. S-palmitoylation of the substrate protein is preceded by the auto-acylation of the ZDHHC enzymes (middle panel). The palmitoyl group is moved from the palmitoyl-CoA molecule onto the cysteine residue of ZDHHC, creating an intermediate state. Then, the fatty acyl group is moved into the thiol group of a cysteine residue of a substrate protein (right panel). The reverse process of removing the palmitoyl group from protein (depalmitoylation) is catalyzed by palmitoyl-protein thioesterases (left panel). The inhibitors of enzymes are shown in purple oval boxes. Abbreviations: NtBuHA, N-(tert-butyl) hydroxylamine; APT1, acyl-protein thioesterase 1; APT2, acyl-protein thioesterase 2; APT1L, APT1-like; PPT1, palmitoyl-protein thioesterase 1; PPT2, palmitoyl-protein thioesterase 2; ZDHHC, zinc finger DHHCs; CoA, coenzyme A; 2-BP, 2-bromopalmitate; CMA, cyano-myracrylamide; C, cysteine residue.

**Figure 2 cells-12-00387-f002:**
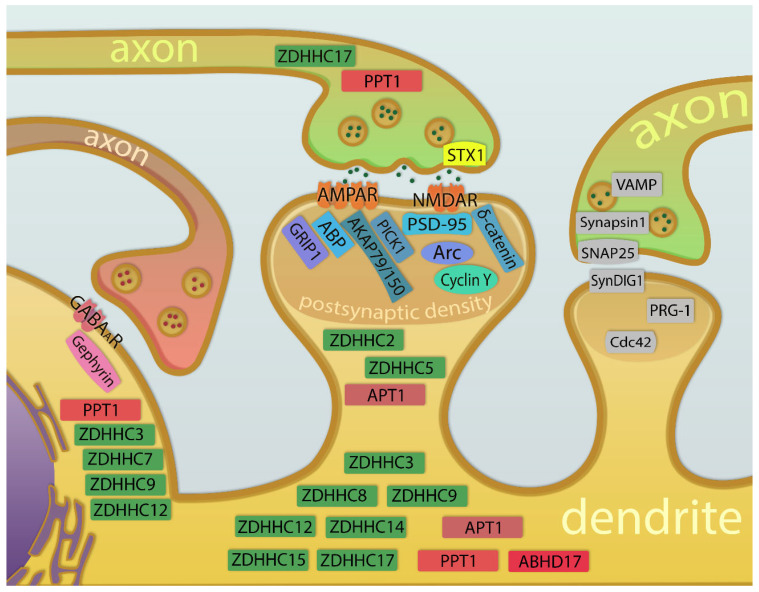
Schematic illustration of the proteins and enzymes of the palmitoylation machinery mentioned in this review in the context of function and plasticity of inhibitory and excitatory synapse. Synaptic proteins in the inhibitory (left) and excitatory (middle) synapses are symbolically depicted, together with palmitoyltransferases (green) and depalmitoylating enzymes (red). The right panel shows proteins (gray) that undergo neuronal activity-dependent palmitoylation and could potentially be involved in the regulation of the function or plasticity of the excitatory synapses (putative candidates not yet confirmed with functional experiments). APT1, acyl-protein thioesterase 1; PPT1, palmitoyl-protein thioesterase 1; ABHD17, alpha/beta hydrolase domain-containing proteins 17; ZDHHC, zinc finger DHHCs; GABA_A_R, GABA_A_ receptor; AMPAR, AMPA receptor; NMDAR, NMDA receptor; PSD-95, Postsynaptic density protein 95; PICK1, protein interacting with C-kinase; AKAP79/150, A-kinase anchoring protein 79/150; ABP, AMPA receptor binding protein; GRIP1, glutamate receptor interacting protein-1; Arc, activity-regulated cytoskeletal-associated protein; VAMP, vesicle-associated membrane protein; SNAP25, synaptosomal-associated protein 25; SynDIG1, synapse differentiation Inducing 1 protein; PRG-1, plasticity-related gene 1 protein; Cdc42, cell division control protein 42 homo.log.

**Table 1 cells-12-00387-t001:** Proteins are primary substrates for 4 classes of lipid modifications (which do not represent the whole spectrum of lipid modification): S-acylation, N-myristoylation, isoprenylation, and the fatty acylation of secreted proteins. Every type of lipid modification is catalyzed by unique enzymes and targets a specific amino acid of substrate protein creating characteristic linkage. Most common form of S-acylation is S-palmitoylation. The unique quality of S-palmitoylation is its reversibility, and most lipid modifications create stable bonds/linkage.

Class of Lipid Modification	Type of Lipid Modification	PTM Character	Lipid	Amino Acid	Linkage
S-acylation	S-palmitoylation	reversible	Palmitate (16:0)	cysteine	thioester
			Stearate (18:0)	cysteine	thioester
			Oleate (18:1)	cysteine	thioester
			Arachidonate (20:4n-6)	cysteine	thioester
			Eicosapentanoate (20:5n-3)	cysteine	thioester
N-myristoylation	N-myristoylation	stable	Myristate (14:0)	N-terminal glycine	amide
Isoprenylation	S-farnesylation	stable	Farnesyl (15:0)	cysteine	thioester
	S-geranylgeranylation	stable	Geranylgeranyl (20:0)	cysteine	thioester
Fatty acylation of secreted proteins by membrane bound O-acyl transferases	N-palmitoylation	stable	Palmitate (16:0)	N-terminal glycine	amide
	O-acylation	stable	Octanoate (8:0)	cysteine	oxyester
		reversible	monounsaturated form of palmitate (palmitoleic 16:1Δ9)	cysteine	oxyester

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
