# Peer review of "S-Palmitoylation of Synaptic Proteins in Neuronal Plasticity in Normal and Pathological Brains"

_cells, 2023, doi:10.3390/cells12030387_

Round 1

Reviewer 1 Report

The narrative review article by Buszka et al. " S-palmitoylation of synaptic proteins in neuronal plasticity in normal and pathological brain", focused on the S-palmitoylation of proteins involved in synaptic and brain functions, offers a good compilation of evidence on the impact of this post-translational modification on the physiology of excitatory and inhibitory synapses as well as on the importance of S- palmitoylation in cognitive functions. The text is acceptably organized and the content summarizes the state of the art in this field and provides neurobiologists with a fairly broad view of this aspect of biology.

In summary, this reviewer recommends the publication of the review Buszka et al. in the special issue "Multitasking Proteins and Their Involvement in Pathogenesis" of the “International Journal of Molecular Sciences” provided the following aspects are answered according to the following comments.

MAJOR POINTS

Some parts of the text are written in an excessively telegraphic way and  should be written in more fluent prose (e.g., the text of lines 25-31).

Chapter 2.1 should be summarized (or even deleted by integrating its content in the preceding paragraph) linking it to S-palmitoylation. Alternatively, the chapter could be devoted to the importance of palmitoylation in synaptic signaling mediated by membrane receptors in general and G protein-coupled receptors (GPCRs) in particular and its importance in plasticity phenomena.

The title of the chapter 2.2 should include the concept of plasticity in relation to palmitoylation/depalmitoylation turnover. Alternatively, chapters 2.2 and 2.3 could be fused.

I recommend reorganizing section 4 into two subsections referred to neurological/neurodegenerative diseases (including epilepsy and Parkinson) and neuropsychiatric disorders.

There are numerous paragraphs in which the authors reproduce almost identically the abstract in the original research articles (e.g., lines 579-583). This sometimes makes the description out of context, apparently contradictory with previous statements and difficult to interpret (e.g., lines 588-596). Authors should make an effort to rewrite and contextualize the results of other authors. Lines 583-587: the synthesis of reference 157 (which the authors must correct) does not seem to conform to what was described in the original article.

MINOR POINTS

References: please, check an unify the style of the bibliographical references in the body of the text (e.g., lines 33, 40, 64, 111, 137, 142, 144, 148, 150, 160, 165, 251, 255, 260, 262, 403, 432, 451, line 112 –“recently reviewed by [32]” – and more…).

Examples of typos. Line 182; remove spaces between “compounds” and “known”. Line 184; remove comma after “acrylamide-based”. Line 355, remove the semicolon after “namely”. Line 257; replace “Shaeffer” with “Schaffer”. Please use the spell checker of your text editor.

Line 193: replace “from” with “of”.

Line 236: replace “GABAARs (GABA type A receptors)” with “GABA type A receptors (GABAARs)”.

Lines 235-238: replace “excitatory postsynaptic potential (EPSP)” and “inhibitory postsynaptic potential (IPSP)” with “excitatory postsynaptic potentials (EPSPs)” and “inhibitory postsynaptic potentials (IPSPs)”.”

Lines 239-240: replace “EPSP and IPSP” with “EPSPs and IPSPs”.

Line 162: give the meaning of the acronym “Gαs”; α subunits of heterotrimeric G proteins (Gαs).

Lines 168-169: Replace “depalmitoylases alpha/beta hydrolase domain-containing proteins 17 (ABHD17) isoforms 168 A/B/C were discovered” with “A, B and C isoforms of α/β-hydrolase domain-containing protein 17 (ABHD17) were identified as responsible for that activity.”

Lines 320-321: The authors mention glycine/NMDA and glycine/bicuculline protocols as inducers of cLTP and cLTD, respectively. A bibliographical reference is required to know which specific experimental paradigms they refer to. For instante, are “glycine cLTP” and the “glycine/NMDA” the same model?

Lines 339-342: the link between palmitoylation of δ-catenin and PRG-1 and the cLTP phenomenon is not clear. Is there a causal relationship or just a coincidence that suggests a causal relationship? please clarify.

Line 406: place “(HFS)” after “stimulation”.

Lines 448-454: state somewhere that SynDIG1 is an AMPAR-interacting transmembrane protein.

Line 495: do the authors mean “given the low activity of PATs” or “under conditions of low PAT activity”?

Line 499: mEPSCs/EPSCs is not an acronym of excitatory synaptic signals. Authors can directly use “LTD of EPSCs”, as the acronym has been defined above.

Line 501: give the meaning of PICK1 acronym.

Line 505: give the meaning for mEPSCs (miniature EPSCs).

Line 507: the term “homeostatic plasticity” is first mentioned here. Add a brief explanation of the concept.

Lines 514-515: the model is not clear. Cultured neurons, slices, animals? Please,specify.

Line 567: remove spaces between “receptors” and “suggests”.

Line 573: give the meaning of PAS acronym.

Line 592: give the meaning of CDKL5

Lines 612-613: are ABHD17 variants called the membrane-anchored serine hydrolases? Please, check.

Lines 612-613: what did Fukata et al. revealed? The enzymes responsible for depalmitoylation of PSD-95? Rewritte.

Line 614: add comma before “i.e.”.

Line 615: check and unify the nomenclature used for NMDAR subunits (GluN2A or NR2A?).

Lines 635-636: the subject of the sentence is not clear. Rewritte.

Lines 643-647: add STX1A and STX1B as separate acronyms syntaxin-1A and 1B. Unify nomenclature (syntaxin 1, syntaxin 1 or syntaxin1?)

Line 665: remove “Sindbis”. “viral expression” is sufficient.

Lines 687-703: the text is confusing. It is not clear what is the effect of palmitoylation at the CLCRC motif of Arc and the role of MEF2 in Arc palmitoylation-mediated plasticity. What do the authors mean with Arc induction?

Lines 703-707: express the same idea in a single shorter sentence.

Line 710: why “adverse”?

Line 718: for clarity, add the word “negative” before the word “regulatory”.

Lines 722-726: rewrite for clarity.

Lines 728-734: the relationship between GluA1 exocytosis/PSD-95 clustering and LTP is unclear. Was LTP reduced upon overexpression of WT cyclin Y or must be inferred by the reader?

Lines 740-741: in what process? Who “regulates some aspects of the process of SVs recycling? Syn1 or ZDHHC5? Please, disambiguate.

Lines 749-752: the explanation given does not support that Syn1 palmitoylation is involved in SV-related presynaptic events unless Syn1 palmitoylation was decreased in ZDHHC5 KO mice. Was it?

Line 764: specify what model was used. Culture, slices…

Line772: what is the rationale of using S-PALM acronym here?

Line 780-781: spell R7BP “as “regulator of G protein signaling (RGS) 7-binding protein”.

Line 817: add the article “a” before “Mg2+”.

Line 818: at what level does 2-BP inhibits ZDHHC17?

Lines 843-847: There is a syntax error or something is missing from the sentence.

Lines 870-872: in what model?. “almost doubled” is not concordant with “ZDHHC9 knockdown”

Lines 872-880: the text is just a series of statements with lack of contextualization. What is the relationship between the observed γ2 subunit of GABAAR cluster density and the increased amplitude of both mEPSC and IPSC? What are the “discrepancies in the electrophysiological observations in vitro and slice recordings? Marcoscopic refers to the sense of sight and not to functional registers.

Line 875: the verb is not concordant with the subject.

Line 882: replace “to have palmitoylated” with “to palmitoylate”.

Line 952: check syntaxis. Is there a comma missing?

Line 956: Glu2B or GluN2B?

Lines 959-960: “primary cortical PPT1 knockout neurons” of “primary cortical neurons of PPT1 knockout mice”?

Line 964: is “implied” correct?

Lines 974-975: check writing.

Lines 979-982: check writing and content. It is not clear what neurotransmission relies on "is regulated by" may be more accurate.

Line 1011: what do the authors mean with “especially in assemblies”?

Lines 1029:1030: what led to impaired fear memory formation, fear conditioning or ZDHHC2 knockdown?. Please, fuse sentences.

Line 1062: “least amounts” is not correct.

Line 1088: rewrite “Mutation of cysteines in protein that is a substrate to palmitoylation is…”. For instance, “Mutation of cysteines in PAT protein substrates constitute…”

Line 1096-1110: the sentence “There is a line of evidence suggesting…” makes no sense here, since this premise is the basis of the entire chapter. In any case, since no direct evidence is provided for the relationship between palmitoylation of the mentioned proteins and memory and learning, I recommend deletion of this paragraph entirely. Maybe the last sentence is an exception to this, but it should somehow be integrated into another part of the discussion.

Lines 1120-1120: do you mean that the learning was delayed and required more test cycles? In any case, learning was affected even if the animals passed the last tests the last tests in a similar way to the controls, which is typical in many cognitive impairment paradigms with repetitive trials. That means that the learning capacity was in fact affected.

Lines 1130-1131: what do you mean with “however the case with palmitoylation is different”? That palmitoylation is increased? Is there any reason to believe that the changes occur in the same direction?.

Line 1135: was palmitoylation increased in aged animals compared to young adults? Clarify.

Lines 1125-1142: this text could be well moved as a second paragraph of chapter 4, adding “aging” before “human neurological disorders” in line 1145. Also it could be linked to dementia and Alzheimer’s disease.

Line 1146: remove “Several reviews provide a great reference of that matter”.

Lines 1147-1148: remove “Below we discuss to what […] functions”

Line 1150: remove comma after “impairment”.

Lines 1151-1153: remove “Some other diseases not described […] proteins and in particular ZDHHC8”. Modify references accordingly.

Lines 1162-1172: compress the description of the clinical manifestations into one sentence, and place it before the sentence beginning with “One class of diseases…”. The last statement may be relevant, but is out of context. Please, try to integrate it somewhere.

Lines 1173-1183: Rewrite avoiding telegraphic style. Try to rewrite in a more fluid style with appropriate links between biochemical/molecular events and pathophysiology/clinical manifestations. The last sentence is meaningless if therapies do not target palmitoylation-related players.

Line 1206: The meaning of GlunN2B has been specified above. NMDAR2B acronym makes no sense. Do you mean surface expression of GluN2B subunit-containing NMDA receptors?

Line 1209: replace “its” with “the”.

Lines 1212-2214: More likely, perturbed function of huntingtin-associated PAT ZDHHC13 in YAC128 mice promotes NMDA toxicity in the striatal neurons, thus increasing their susceptibility to apoptosis and contributing to HD pathology.

Line 1216: Although the meaning of APT (acyl-protein thioesterase) has been given before for specific isoforms, I recommend giving it here again for generic APTs.

Line 1219: why alternative model?. Isn’t it just another transgenic HD mouse model?

Line 1220: remove the commas flanking “ML348”.

Line 1121: “and to ameliorate”.

Line 1124: do you mean cortical neurons derived from iPSCs of HD patients?

Lines 1251-1253:  remove the sentence “Finally, ZDHHC9 mutant mouse…” and discuss the findings in the animal model available for several years.

Lines 1288-1289: The meaning for ABHD17 in not necessary as it has been given above.

Author Response

We thank all Reviewers for suggestions that will further enhance our review article. In this regard, the questions or concerns raised by Reviewers were thoroughly addressed and the appropriate corrections in the revised manuscript have been made accordingly. The respective changes made in the text of the manuscript were highlighted in bold and additionally specified below.

Reviewer 1 Comments and Suggestions for Authors

The narrative review article by Buszka et al. " S-palmitoylation of synaptic proteins in neuronal plasticity in normal and pathological brain", focused on the S-palmitoylation of proteins involved in synaptic and brain functions, offers a good compilation of evidence on the impact of this post-translational modification on the physiology of excitatory and inhibitory synapses as well as on the importance of S- palmitoylation in cognitive functions. The text is acceptably organized and the content summarizes the state of the art in this field and provides neurobiologists with a fairly broad view of this aspect of biology.

In summary, this reviewer recommends the publication of the review Buszka et al. in the special issue "Multitasking Proteins and Their Involvement in Pathogenesis" of the “International Journal of Molecular Sciences” provided the following aspects are answered according to the following comments.

MAJOR POINTS

Reviewer 1: Some parts of the text are written in an excessively telegraphic way and  should be written in more fluent prose (e.g., the text of lines 25-31).

Authors response: We thank the Reviewer for this comment. Whenever possible, we have rewritten the text. We have also made language editing to improve the quality of the text as English is not the mother-tongue of authors of this manuscript.

Reviewer 1: Chapter 2.1 should be summarized (or even deleted by integrating its content in the preceding paragraph) linking it to S-palmitoylation. Alternatively, the chapter could be devoted to the importance of palmitoylation in synaptic signaling mediated by membrane receptors in general and G protein-coupled receptors (GPCRs) in particular and its importance in plasticity phenomena.

Authors response: The aim of chapter 2.1 was to introduce the key ionotropic receptors involved in synaptic transmission and explain concept of synaptic plasticity to the reader, as both topics are extensively discussed in the following chapters in the context of S-palmitoylation. In our view this chapter fits “Palmitoylation – the role in the function and plasticity of synapses” chapter rather than Introduction. Therefore we propose to leave this chapter as is. The role of S-palmitoylation in the function of 9 classes of GPCRs has been recently extensively reviewed by Naumenko and Ponimaskin (2018, Neural plasticity Journal). In addition in another Review by Ji and Skup 2021 palmitoylation, GPCRs and structural plasticity has been described. In addition, unlike ionotropic receptors, the link for palmitoylation and GPCRs in the context of modified synaptic function or plasticity is largely lacking and is not extensively discussed in our manuscript. Therefore, we propose to mention GPCRs link with palmitoylation and cite mentioned reviews in chapter Conclusions.

Reviewer 1: The title of the chapter 2.2 should include the concept of plasticity in relation to palmitoylation/depalmitoylation turnover. Alternatively, chapters 2.2 and 2.3 could be fused.

Authors response: We agree with the Reviewer. We have fused chapters 2.2 and 2.3 in the revised version of the manuscript and renamed this chapter “Neuronal activity and synaptic plasticity affect the palmitoylation/depalmitoylation cycles”.

Reviewer 1: I recommend reorganizing section 4 into two subsections referred to neurological/neurodegenerative diseases (including epilepsy and Parkinson) and neuropsychiatric disorders.

Authors response: We agree with the Reviewer. We have divided chapter 4 accordingly and include now sections: neurodegenerative diseases (Alzheimer’s disease, Huntington’s disease, Neuronal ceroid lipofuscinoses), neuropsychiatric disorders (Major Depressive Disorder, Schizophrenia) and neurological disorders (i.e. Chromosome X-linked intellectual disability).

Reviewer 1: There are numerous paragraphs in which the authors reproduce almost identically the abstract in the original research articles (e.g., lines 579-583). This sometimes makes the description out of context, apparently contradictory with previous statements and difficult to interpret (e.g., lines 588-596). Authors should make an effort to rewrite and contextualize the results of other authors. Lines 583-587: the synthesis of reference 157 (which the authors must correct) does not seem to conform to what was described in the original article.

Authors response: We agree with the Reviewer. We have rewritten the manuscript to better contextualize results of other authors whenever possible. With regard to citation 157, in the revised version of the manuscript, we wrote more precisely: “Indeed, PSD-95 is palmitoylated at two cysteine residues (C3 and C5) at its N-terminus, and palmitoylation at this locus is required for clustering of PSD-95 (El Husseini et al., 2002). In addition, neurons expressing C3 and C5 PSD-95 mutant exhibit impaired AMPARs surface availability and reduced EPSCs (Schnell et al., 2002)”. In addition, with regard to citation 157,  we previously wrote “Cells with PSD-95 mutants lacking cysteines 3 and 5 do not undergo neuronal-activity-dependent changes in PSD-95 clusterization.”. We have removed that sentence for two reasons. First, it is not precise. In this study, authors showed the impact of glutamate on surface availability of GluR4 in palmitoylation-deficient prenylated PSD-95. Second, this work does not fit the paragraph which describes ZDHHCs that palmitoylate PSD-95.

Reviewer 1 MINOR POINTS:

References: please, check an unify the style of the bibliographical references in the body of the text (e.g., lines 33, 40, 64, 111, 137, 142, 144, 148, 150, 160, 165, 251, 255, 260, 262, 403, 432, 451, line 112 –“recently reviewed by [32]” – and more…).

Authors response: All cited references have been unified in the text. For example: [4], [5] have been changed to [4, 5] as per the journal guidelines. The word ‘reviewed by’ has been deleted from the citations.

Examples of typos. Line 182; remove spaces between “compounds” and “known”. Line 184; remove comma after “acrylamide-based”. Line 355, remove the semicolon after “namely”. Line 257; replace “Shaeffer” with “Schaffer”. Please use the spell checker of your text editor.

Authors response: We thank Reviewer for noting that. All the suggested words have been corrected.

Lines 320-321: The authors mention glycine/NMDA and glycine/bicuculline protocols as inducers of cLTP and cLTD, respectively. A bibliographical reference is required to know which specific experimental paradigms they refer to. For instante, are “glycine cLTP” and the “glycine/NMDA” the same model?

Authors response: We agree with the Reviewer. In the revised version of the manuscript we explain protocols more precisely: “One commonly used in vitro model of inducing neuronal plasticity involves glycine/bicuculline which chemically promotes synaptic LTP by selective activation of synaptic NMDARs (Lu et al, 2000 Neuron; Brigidi et al., 2015). In contrast to chemical LTP, chemical LTD induced with glycine/NMDA activates both synaptic and extrasynaptic NMDARs, resulting in AMPAR internalization and synaptic depression (Musleh et al. 1996, Li et al., 2011).

Lines 339-342: the link between palmitoylation of δ-catenin and PRG-1 and the cLTP phenomenon is not clear. Is there a causal relationship or just a coincidence that suggests a causal relationship? please clarify.

Authors response: With regard to δ-catenin, it is a substrate to ZDHHC5. In addition, both proteins are upregulated following cLTP and translocated together from spine to dendritic shaft. shRNA against ZDHHC5 prevents co-localization of δ-catenin with PSD95 induced with cLTP (Brigidi et al., 2015 Nat Comm). Thus, it seems that both ZDHHC5 and cLTP are crucial for δ-catenin palmitoylation of localization in neurons. With regard to PRG-1, palmitoylation of this protein was found to occur both after cLTP in vitro and in vivo following learning in fear conditioning paradigm (Nasseri et al., 2022 Science Signaling). While there are many articles linking δ-catenin palmitoylation with synaptic plasticity and learning, there is only one linking PRG-1 protein in the same way. Still however, evidence are indirect and we have emphasized that in the original version of the text (page 24): “Although δ-catenin palmitoylation deficient animals were not tested for cognitive impairments, it is possible that in vivo synapse plasticity associated with learning events requires palmitoylation of that protein [30]. Similarly, it was proposed based on in vitro studies that stability and subcellular localization of SynDIG1 and indirectly AMPAR may be regulated by activity-dependent palmitoylation [155].”

Line 495: do the authors mean “given the low activity of PATs” or “under conditions of low PAT activity”?

Authors response: We have rephrased the sentence: “Thus, under conditions of low PAT activity, the effect of 2-BP would be less extensive compared to NtBuHA, while it may be pronounced in pathological conditions.”

Line 507: the term “homeostatic plasticity” is first mentioned here. Add a brief explanation of the concept.

Authors response: We agree with the Reviewer. In the revised version of the manuscript it now reads: “Pharmacological manipulation of protein palmitoylation was shown to effectively block homeostatic plasticity, which is defined broadly as a set of neuronal changes that restore activity to a setpoint following perturbation.”

Lines 514-515: the model is not clear. Cultured neurons, slices, animals? Please,specify.

Authors response: The model was cultured hippocampal neurons. It is now explained in the revised version of the manuscript.

Lines 612-613: are ABHD17 variants called the membrane-anchored serine hydrolases? Please, check.

Authors response: This description of ABHD17 proteins is not official nomenclature (EC 3.1.2.22  palmitoyl[protein] hydrolase) but descriptive way to name enzymes followed after original article by Fukata and coworkers. Serine hydrolases represent a large family of >200 enzymes both found in the cytoplasm and membranes. In the revised version of the manuscript we now use a description “serine hydrolases”: “Fukata and coworkers identified the members of the ABHD17 serine hydrolases as the PSD-95 depalmitoylating enzymes.”

Lines 612-613: what did Fukata et al. revealed? The enzymes responsible for depalmitoylation of PSD-95? Rewrite.

Authors response: We have rewritten this sentence and now it reads: “Fukata and coworkers identified the members of the ABHD17 serine hydrolases as the PSD-95 depalmitoylating enzymes.”

Lines 635-636: the subject of the sentence is not clear. Rewrite.

Authors response: We agree with the Reviewer. We have rephrased the sentence which now reads: “AKAPCS mutant did not exhibit altered presynaptic release probability. Thus, LTP deficits in AKAPCS mice were specifically related to impaired regulation of AMPARs permeability to Ca2+ ions.”

Lines 687-703: the text is confusing. It is not clear what is the effect of palmitoylation at the CLCRC motif of Arc and the role of MEF2 in Arc palmitoylation-mediated plasticity. What do the authors mean with Arc induction?

Authors response: We thank Reviewer for noting the part of the text which may be unclear. We have rephrased that paragraph to sound more clear. Briefly, we explained that Arc expression is induced experimentally by acute expression of constitutively active MEF2 (biolistic transfection of organotypic slice cultures). Also we explain, that palmitoylated Arc is necessary for MEF2-mediated depression of synaptic transmission since Arc non-palmitoylatable mutant shows not MEF2-induced changes in synaptic transmission.

In the revised version of the manuscript we wrote: “In wild-type slice cultures of the hippocampus, acute expression of constitutively active transcription factor MEF2 achieved via biolistic transfection induces Arc expression and results in a negative modulatory effect on AMPAR-mediated synaptic transmission manifested by a decreased frequency of mEPSCs. In this model, MEF2-induced depression of mEPSC frequency observed in the presence of palmitoylated Arc was rescued by co-transfection with the non-palmitoylatable Arc mutant protein. Thus, Arc palmitoylation may be essential in some forms of the plasticity at excitatory synapses.”

Lines 722-726: rewrite for clarity.

Authors response: We agree with the Reviewer. The new rewritten sentence reads: “Overexpression of WT cyclin Y in hippocampal neurons leads to decreased amplitudes of mEPSC and reduced presence of GluA1 on the surface of the spine while overexpression of palmitoylation-deficient (or myristoylation-deficient) mutants has exactly the opposite effect. Spine shape and the synaptic expression of PSD-95 is also differentially modulated by cyclin Y. In particular, overexpression of wild-type cyclin Y negatively affects those parameters while expression of the mutant leads to a drastic enhancement in the size of a spine and the presence of PSD-95 at the synaptic membrane.”

Line 710: why “adverse”?

Authors response: “Adverse” was used to emphasize Cyclin Y effect on the AMPARs delivery and functional changes in the synapse. In the revised version on the manuscript we have rephrased that sentence which now reads:” Cyclin Y in neurons is located mainly within dendritic spines where it inhibits plasticity-induced AMPA receptor delivery to synapses and LTP in the hippocampus.”

Lines 728-734: the relationship between GluA1 exocytosis/PSD-95 clustering and LTP is unclear. Was LTP reduced upon overexpression of WT cyclin Y or must be inferred by the reader?

Authors response: In the original manuscript we explained that “Overexpression of WT cyclin Y in hippocampal neurons leads to decreased amplitudes of mEPSC and reduced presence of GluA1 on the surface of the spine while overexpression of palmitoylation-deficient (or myristoylation-deficient) mutants has exactly the opposite effect”. We then explained that “Under glycine LTP, neurons overexpressing WT cyclin Y showed decreased exocytosis of GluA1 and clustering of PSD-95 compared to the control, while opposite effects were observed in neurons overexpressing a palmitoylation-deficient mutant of cyclin Y”. We further wrote “LTP at the Schaffer collateral-CA1 synapses is negatively affected by overexpression of WT cyclin Y in CA1 neurons”.  In other words yes, cyclin Y expression is inversely related to the magnitude of LTP and this could be directly inferred from the text.

Lines 740-741: in what process? Who “regulates some aspects of the process of SVs recycling? Syn1 or ZDHHC5? Please, disambiguate.

Authors response: This sentence has been corrected to indicate Synapsin-1.

Lines 749-752: the explanation given does not support that Syn1 palmitoylation is involved in SV-related presynaptic events unless Syn1 palmitoylation was decreased in ZDHHC5 KO mice. Was it?

Authors response: The cited article provides evidence for palmitoylated synapsin-1 role in presynaptic release machinery organization. This article provides electron microscopy data supporting the role of palmitoylated synapsin-1 in organization of synaptic vesicles and their number. It also provides functional readout – non palmitoylated synapsin-1 neurons in vitro had impaired synaptic vesicle recycling in response to 55mM KCL as visualized in time with membrane dye FM4-64. Thus, we think that this article supports a view that Synapsin-1 palmitoylation is involved in SV-related presynaptic events. With regards to ZDHHC5 authors provide two links. First, co-expression of synapsin-1 with ZDHHC5 results in increased palmitoylated form of synapsin-1. The opposite was observed in ZDHHC5 KO animals. Second, ZDHHC5 KO animals had the same phenotype in terms of changes in vesicle number and synaptic vesicle organization as synapsin-1 palmitoylation deficient mutant in vitro. In the revised version of the manuscript we wrote:” Final piece of evidence supporting the importance of Syn1 palmitoylation in SV-related presynaptic events comes from the ZDHHC5 knockout mice. In this model, decreased palmitoylation of Syn1 was observed while electron microscopy revealed defects in the SVs clustering manifested by decreased density and number of SVs in the presynaptic compartment”

Line772: what is the rationale of using S-PALM acronym here?

Authors response: We thank reviewer for noting this mistake. We have corrected S-PALM to simply palmitoylation.

Line 818: at what level does 2-BP inhibits ZDHHC17?

Authors response: We are not aware of a single pharmacological study addressing specifically IC50 of 2BP for ZDHHC17. It is generally acknowledged that IC50 for most palmitoyltransferases is in the range of 10-15µM ( reviewed by Zheng and coworkers, 2013, J. Am. Chem. Soc.). With respect to studies cited in our review authors reported that in oocytes, where overexpression of ZDHHC3 or ZDHHC17 was obtained, 75 µm of 2-bromopalmitate (2BP) applied for 3h reduced Ca2+ transport by 80% and Mg2+ transport by 50% for ZDHHC3 and ZDHHC17, respectively.

Lines 872-880: the text is just a series of statements with lack of contextualization. What is the relationship between the observed γ2 subunit of GABAAR cluster density and the increased amplitude of both mEPSC and IPSC? What are the “discrepancies in the electrophysiological observations in vitro and slice recordings? Marcoscopic refers to the sense of sight and not to functional registers.

Authors response: We thank Reviewer for raising that point. The discrepancies mentioned in the manuscript refer to data obtained by authors. First knockout of ZDHH9 in mice reduced the frequency of IPSCs but knockdown of ZDHHC9 in vitro did not change this parameter. Second, less complex dendritic arbors and decreased gephyrin and γ2 subunit of GABARs cluster density is expected to result in weaker GABAergic synaptic transmission. However, this resulted in increased amplitude of IPSCs without any change in synaptic events frequency. In the revised version of the text we have removed adjective “macroscopic”. We have also rephrased the paragraph which now reads: “While it is difficult to unequivocally interpret morphological data in the context of electrophysiological observations in vitro and in slice recordings, ZDHHC9 seems to be essential for supporting network formation and maintaining excitatory to inhibitory synaptic balance. Indeed, ZDHHC9 knockout mice exhibited more and longer seizure-like events as visualized with electroencephalographic recordings [36].”

Line 1011: what do the authors mean with “especially in assemblies”?

Authors response: We have deleted this unnecessary information.

Lines 1029:1030: what led to impaired fear memory formation, fear conditioning or ZDHHC2 knockdown?. Please, fuse sentences.

Authors response: We thank Reviewer for noting that. ZDHHC2 knockdown impaired fear memory formation.

Line 1088: rewrite “Mutation of cysteines in protein that is a substrate to palmitoylation is…”. For instance, “Mutation of cysteines in PAT protein substrates constitute…”

Authors response: As requested by Reviewer 2 we have removed that sentence from paragraph as it was redundant with regard to chapter 2.

Line 1096-1110: the sentence “There is a line of evidence suggesting…” makes no sense here, since this premise is the basis of the entire chapter. In any case, since no direct evidence is provided for the relationship between palmitoylation of the mentioned proteins and memory and learning, I recommend deletion of this paragraph entirely. Maybe the last sentence is an exception to this, but it should somehow be integrated into another part of the discussion.

Authors response: We agree with the Reviewer. Although palmitoylation has been implied in learning and memory for two decades, it was just last year when a first paper was published fully experimentally addressing this issue. In this chapter we tried to put everything that could was published in this topic. However, we agree that some links between palmitoylation and learning and memory are indirect. Therefore, we have removed that part of the text from the paragraph where indirect link between δ-catenin, palmitoylation and learning was described.

Lines 1120-1120: do you mean that the learning was delayed and required more test cycles? In any case, learning was affected even if the animals passed the last tests the last tests in a similar way to the controls, which is typical in many cognitive impairment paradigms with repetitive trials. That means that the learning capacity was in fact affected.

Authors response: According to original paper and authors interpretation of data, Cyclin Y could only affect the speed but not the capacity of learning.

Lines 1130-1131: what do you mean with “however the case with palmitoylation is different”? That palmitoylation is increased? Is there any reason to believe that the changes occur in the same direction?. Was palmitoylation increased in aged animals compared to young adults? Clarify.

Authors response: We thank the Reviewer for that comment. We have rephrased that sentence which now reads: “In the mouse brain, total protein expression of NMDAR subunits was previously reported to decrease with age [219] in contrast to palmitoylation levels of these receptors”. As we wrote in the text, “It was found that in aging ….there was an increase in the levels of palmitoylation of GluN2B, GluN2A, and Fyn in the prefrontal cortex which most likely resulted from a disturbed cellular palmitoylation cycle and had detrimental effects on reference memory”. We also wrote “aging led to an increase in palmitoylation of APT1 in the prefrontal cortex”.

Lines 1125-1142: this text could be well moved as a second paragraph of chapter 4, adding “aging” before “human neurological disorders” in line 1145. Also it could be linked to dementia and Alzheimer’s disease.

Authors response: We agree with the Reviewer and have moved the paragraph to Chapter 4 in the revised version of the manuscript.

Lines 1162-1172: compress the description of the clinical manifestations into one sentence, and place it before the sentence beginning with “One class of diseases…”. The last statement may be relevant, but is out of context. Please, try to integrate it somewhere.

Authors response: Agreed and corrected.

Lines 1173-1183: Rewrite avoiding telegraphic style. Try to rewrite in a more fluid style with appropriate links between biochemical/molecular events and pathophysiology/clinical manifestations. The last sentence is meaningless if therapies do not target palmitoylation-related players.

Authors response: Agreed and corrected. We propose to leave the information on the therapies. Currently, they involve for instance intracerebroventricularly administered PPT1 enzyme replacement therapy (recombinant enzymes) and gene therapies involving introduction of healthy NCL gene.

Lines 1251-1253:  remove the sentence “Finally, ZDHHC9 mutant mouse…” and discuss the findings in the animal model available for several years.

Authors response: It is not clear to us what exactly should be changed. X-Linked intellectual disability covers over 150 syndromes and is associated with mutations to >100 genes. Does the Reviewer refer to Fragile X syndrome ? We only mention that ZDHHC9 mutant mouse line may be just another interesting model system.

In addition all the following minor comments od Reviewer 1 were grouped and addressed in the same way:

Line 499: mEPSCs/EPSCs is not an acronym of excitatory synaptic signals. Authors can directly use “LTD of EPSCs”, as the acronym has been defined above.

Line 406: place “(HFS)” after “stimulation”.

Lines 448-454: state somewhere that SynDIG1 is an AMPAR-interacting transmembrane protein.

Line 193: replace “from” with “of”.

Line 236: replace “GABAARs (GABA type A receptors)” with “GABA type A receptors (GABAARs)”.

Lines 235-238: replace “excitatory postsynaptic potential (EPSP)” and “inhibitory postsynaptic potential (IPSP)” with “excitatory postsynaptic potentials (EPSPs)” and “inhibitory postsynaptic potentials (IPSPs)”.”

Lines 239-240: replace “EPSP and IPSP” with “EPSPs and IPSPs”.

Line 162: give the meaning of the acronym “Gαs”; α subunits of heterotrimeric G proteins (Gαs).

Lines 168-169: Replace “depalmitoylases alpha/beta hydrolase domain-containing proteins 17 (ABHD17) isoforms 168 A/B/C were discovered” with “A, B and C isoforms of α/β-hydrolase domain-containing protein 17 (ABHD17) were identified as responsible for that activity.”

Line 501: give the meaning of PICK1 acronym.

Line 505: give the meaning for mEPSCs (miniature EPSCs).

Line 567: remove spaces between “receptors” and “suggests”.

Line 573: give the meaning of PAS acronym.

Line 592: give the meaning of CDKL5

Line 614: add comma before “i.e.”.

Line 615: check and unify the nomenclature used for NMDAR subunits (GluN2A or NR2A?).

Lines 643-647: add STX1A and STX1B as separate acronyms syntaxin-1A and 1B. Unify nomenclature (syntaxin 1, syntaxin 1 or syntaxin1?)

Line 665: remove “Sindbis”. “viral expression” is sufficient.

Line 718: for clarity, add the word “negative” before the word “regulatory”.

Lines 703-707: express the same idea in a single shorter sentence.

Line 764: specify what model was used. Culture, slices…

Line 780-781: spell R7BP “as “regulator of G protein signaling (RGS) 7-binding protein”.

Line 817: add the article “a” before “Mg2+”.

Lines 843-847: There is a syntax error or something is missing from the sentence.

Lines 870-872: in what model?. “almost doubled” is not concordant with “ZDHHC9 knockdown”

Line 875: the verb is not concordant with the subject.

Line 882: replace “to have palmitoylated” with “to palmitoylate”.

Line 952: check syntaxis. Is there a comma missing?

Line 956: Glu2B or GluN2B?

Lines 959-960: “primary cortical PPT1 knockout neurons” of “primary cortical neurons of PPT1 knockout mice”?

Line 964: is “implied” correct?

Lines 974-975: check writing.

Lines 979-982: check writing and content. It is not clear what neurotransmission relies on "is regulated by" may be more accurate.

Line 1062: “least amounts” is not correct.

Line 1146: remove “Several reviews provide a great reference of that matter”.

Lines 1147-1148: remove “Below we discuss to what […] functions”

Line 1150: remove comma after “impairment”.

Lines 1151-1153: remove “Some other diseases not described […] proteins and in particular ZDHHC8”. Modify references accordingly.

Line 1206: The meaning of GlunN2B has been specified above. NMDAR2B acronym makes no sense. Do you mean surface expression of GluN2B subunit-containing NMDA receptors?

Line 1209: replace “its” with “the”.

Lines 1212-2214: More likely, perturbed function of huntingtin-associated PAT ZDHHC13 in YAC128 mice promotes NMDA toxicity in the striatal neurons, thus increasing their susceptibility to apoptosis and contributing to HD pathology.

Line 1216: Although the meaning of APT (acyl-protein thioesterase) has been given before for specific isoforms, I recommend giving it here again for generic APTs.

Line 1219: why alternative model?. Isn’t it just another transgenic HD mouse model?

Line 1220: remove the commas flanking “ML348”.

Line 1121: “and to ameliorate”.

Line 1124: do you mean cortical neurons derived from iPSCs of HD patients?

Lines 1288-1289: The meaning for ABHD17 in not necessary as it has been given above.

Authors response: We have corrected the text as requested by the Reviewer1. Corrections are highlighted in bold in the revised version of the manuscript.

Reviewer 2 Report

In the submitted manuscript Buszka and colleagues provide the detailed overview of the role of protein S-palmitoylation in neuronal cells. The review is very comprehensive and successfully summarize the current state of knowledge in regard to palmitoylation processes within CNS. The manuscript is clearly written and well structured. In the first section Authors introduce enzymes catalysing protein S-palmitoylation and depalmitoylation reactions, and chemical compounds used to manipulate S-palmitoylation levels. Next section is devoted to the role of S-palmitoylation status of particular synaptic proteins and its effect on synaptic plasticity in various neuronal plasticity models. The final chapters describe the importance of neuronal proteins S-palmitoylation in different learning and memory test and disturbances in S- palmitoylation observed in and possibly contributing to the development of neurodegenerative disorders and neuropsychiatric conditions. The reference list is exhaustive, and Authors avoid redundancy in meticulous description of topics which have been recently reviewed by other research groups.

However, there are some minor issues to be addressed:  

1.       Figures definitely require some processing. The fonts are not always visible, larger font/darker colours should be used. Figure 2 – it is unclear why there are two excitatory (?) spines pictured here, the one located to the right without any receptors? What is the reason for dividing S-palmitoylated proteins present in excitatory synapses into 2 groups?

2.       Whenever a protein is mentioned for the first time, the full name need to be provided, like for PRG-1, GaS, SNAP, GRIP, TC-10 and so on.

3.       In the section 2 Authors jump from one site of synapse to another. For the sake of logical chapter organization, information on presynaptic proteins should be grouped together and moved to the end of the chapter.

4.       What actually means that S-palmitoylation is “discrete”? Could Authors elaborate more on this definition? Which PTMs are discrete and which of them are not?

5.       In the abstract section there is another puzzling sentence: “Unlike somatic tissue, neural tissue comprises significantly more proteins that are regulated by S-palmitoylation.”. Technically term “somatic tissue/cell” refers to any non-germ line tissue/cell, therefore neural tissue IS somatic tissue. Moreover, there is no experimental evidence to substantiate above statement, e.g. there is no precise measurement data showing that neural tissue (neuronal cells ?) contains more palmitoylated proteins than any other tissue. There are some data on palmitate content provided in lines 57-63, but they don’t determine whether neural tissue/neurons are richer in palmitoylated proteins than any other tissue/cells.

6.       Line 28: there are more than 300 types of PTMs in cell, not 300 PTMs.

7.       Lines 31-33: it is hard to agree that acylation and oxidation based modification is more complex than SUMOylation or polyubiquitination…  please rearrange the sentence.

8.       Line 69: every type of lipid modification is catalysed by (instead of “has its”) specific enzymes (unique is not precise here since enzymes actions may be redundant)

9.       Figure 1 should be referred to when PATs mode of action is described (line 88 and further)

10.   Lines 103-105: S-PATs are recruited to the cell membrane (and not to the cell surface), they are transmembrane proteins with function inside the cell.

11.   Lines 156-163: Are there two distinct groups of depalmitoylating enzymes: PPTs bound to lipid membranes and APTs which are cytosolic? PPTs location is not clearly defined. Is it known?

12.   Lines 199-202: “In contrast” rather than “in addition”, since NtBuHA mimics action of PPTs/ATPs so net effect of this compound on palmitoylation level is opposite to that obtained when inhibitors are used.

13.   Lines 310-314: PSD-95 is depalmitoylated in a ZDHHC2 dependent manner, and ZDHHC2 is PAT, not thioesterase. Could Authors explain ZDHHC2 involvement? Is 2-BP specific biosensor for palmitoylated PSD-95?

14.   Line 385: in small … instead of “as small…”?

15.   Line 956: GluN2b instead of Glu2b

16.   Line 1040: loss-of-function mutation is correct definition

17.   Lines 1096 - 1097: The example of such a protein (GluA1) is given in the previous passage, therefore this intro is irrelevant/should be placed earlier.

18.   Line 1139: between young and old mice

19.   Lines 1242-1243: It is debatable whether the mutations in ZDHHC9 gene are more prevalent in males, or their effects are rather more promptly manifested due to the lack of the backup X chromosome.

20.   Lines 1276-1278: Looks like BACE1 is palmitoylated at 4 cysteine residues and subsequently becomes a substrate for ZDHHC3 etc. Is it true?

21.   Lines 1361-1362: ZDHHC8 simply codes for PAT, instead “which has a significant function of PAT production”

Author Response

We thank all Reviewers for suggestions that will further enhance our review article. In this regard, the questions or concerns raised by Reviewers were thoroughly addressed and the appropriate corrections in the revised manuscript have been made accordingly. The respective changes made in the text of the manuscript were highlighted in bold and additionally specified below.

Reviewer 2 Comments and Suggestions for Authors

In the submitted manuscript Buszka and colleagues provide the detailed overview of the role of protein S-palmitoylation in neuronal cells. The review is very comprehensive and successfully summarize the current state of knowledge in regard to palmitoylation processes within CNS. The manuscript is clearly written and well structured. In the first section Authors introduce enzymes catalysing protein S-palmitoylation and depalmitoylation reactions, and chemical compounds used to manipulate S-palmitoylation levels. Next section is devoted to the role of S-palmitoylation status of particular synaptic proteins and its effect on synaptic plasticity in various neuronal plasticity models. The final chapters describe the importance of neuronal proteins S-palmitoylation in different learning and memory test and disturbances in S- palmitoylation observed in and possibly contributing to the development of neurodegenerative disorders and neuropsychiatric conditions. The reference list is exhaustive, and Authors avoid redundancy in meticulous description of topics which have been recently reviewed by other research groups.

However, there are some minor issues to be addressed: 

Reviewer 2: Figures definitely require some processing. The fonts are not always visible, larger font/darker colours should be used. Figure 2 – it is unclear why there are two excitatory (?) spines pictured here, the one located to the right without any receptors? What is the reason for dividing S-palmitoylated proteins present in excitatory synapses into 2 groups?

Authors response: We thank the Reviewer for comments. We have significantly improved the quality and visibility of figure content. We also explain in the Figure 2 legend that second dendritic spine depicts proteins (shown in gray) that undergo neuronal activity-dependent palmitoylation and could potentially be involved in the regulation of the function or plasticity of the excitatory synapses. However, these are putative likely candidates not yet confirmed with functional experiments unlike proteins depicted in the middle panel.

Reviewer 2: In the section 2 Authors jump from one site of synapse to another. For the sake of logical chapter organization, information on presynaptic proteins should be grouped together and moved to the end of the chapter.

Authors response: We agree with the Reviewer that change in the organization of text make is easier to follow for reader, so we decided to put all information according to the scheme: excitatory synapse – postsynaptic part, excitatory synapse – presynaptic part; inhibitory synapse.

Reviewer 2: What actually means that S-palmitoylation is “discrete”? Could Authors elaborate more on this definition? Which PTMs are discrete and which of them are not?

Authors response: We wanted to emphasize the fact that although S-palmitoylation is speculated to be a common modification (as 41% of synaptic proteins are theoretically palmitoylated [63]) yet position of the palmitoylated cysteine, and the number of cysteines undergoes some form of regulation in the cell. In addition, although many ZDHHCs exhibit redundancy in substrate recognition, evidence exist indicating proteins selectively palmitoylated by single ZDHHCs and at single cysteine (i.e. Yokoi et al., 2016 J Neurosci). This indicates that on the level of individual protein and its primary structure, palmitoylation may be highly regulated and thus discrete.

Reviewer 2: In the abstract section there is another puzzling sentence: “Unlike somatic tissue, neural tissue comprises significantly more proteins that are regulated by S-palmitoylation.”. Technically term “somatic tissue/cell” refers to any non-germ line tissue/cell, therefore neural tissue IS somatic tissue. Moreover, there is no experimental evidence to substantiate above statement, e.g. there is no precise measurement data showing that neural tissue (neuronal cells ?) contains more palmitoylated proteins than any other tissue. There are some data on palmitate content provided in lines 57-63, but they don’t determine whether neural tissue/neurons are richer in palmitoylated proteins than any other tissue/cells.

Authors response: We agree with the Reviewer that the term somatic tissue was misused and we have corrected that. What we wanted to highlight is the fact how profound is the participation of S-palmitoylation as the PTM in synaptic proteins. Therefore we changed this sentence to “Neural tissue is particularly rich in proteins that are regulated by S-palmitoylation”. As we also write on page 8: “Proteomic data analysis indicates that nearly 10% of the human genome codes a form of protein that can be palmitoylated, out of which 41% are synaptic proteins [63]”. Indeed, in our mass spectrometry data (Zareba-Koziol et al., 2019, MCP and unpublished data) we typically detect 30-40% of palmitoylated proteins in synaptoneurosomes from hippocampus in palmitoylated state.

Reviewer 2: Lines 156-163: Are there two distinct groups of depalmitoylating enzymes: PPTs bound to lipid membranes and APTs which are cytosolic? PPTs location is not clearly defined. Is it known?

Authors response: All depalmitoylating enzymes belong to a larger family of serine hydrolases which comprise both membrane bound and cytoplasmic enzymes. However, neither APTs nor PPT1 seems to be membrane bound. In addition to APTs and PPTs, also ABHD hydrolases were described in the text. Subcellular location of APTs and PPTs in neurons has not been systematically addressed to the best of our knowledge. With regard to PPT1, both UniProt database and articles we cite in the manuscript indicate localization in the lysosomes. In addition, Carcel-Trullols et al., (2015 BBA) mention that the PPT1 localization is lysosomal matrix, extralysosomal vesicles, lipid rafts, ER and presynaptic areas in neurons. However, it should be noted that some localizations may be generalized for all tissues thus precise neuronal localization remains unclear.

Reviewer 2: Lines 310-314: PSD-95 is depalmitoylated in a ZDHHC2 dependent manner, and ZDHHC2 is PAT, not thioesterase. Could Authors explain ZDHHC2 involvement? Is 2-BP specific biosensor for palmitoylated PSD-95?

Authors response: We thank Reviewer for the comment. There was a mistake in the text and we have corrected it. The specific biosensor described in the text is a conformation-specific recombinant antibody against palmitoylated PSD-95 created in the laboratory of Masaki Fukata. 2-bromopalmitate was used in this experiment to block newly occurring palmitoylation. Two ZDHHCs were tested and it was found that ZDHHC2 but not ZDHHC3 were responsible for continuous palmitoylation of synaptic PSD95. In the revised version of the manuscript we wrote: “This process was found to be ZDHHC2-dependent and involve PSD-95 translocation away from the postsynaptic membrane as visualized with a PSD-95 palmitoylation-specific biosensor (conformation-specific recombinant antibody). By application of 2-BP which blocks the newly occurring palmitoylation it was found that in basal conditions, almost the entire population of palmitoylated PSD-95 (90%) in young primary neuronal culture, unlike in older cultures, participated in the dynamic palmitoylation cycles.”

Reviewer 2: Lines 1096 - 1097: The example of such a protein (GluA1) is given in the previous passage, therefore this intro is irrelevant/should be placed earlier.

Authors response: We agree with the Reviewer. The sentence has been removed.

Reviewer 2: Lines 1242-1243: It is debatable whether the mutations in ZDHHC9 gene are more prevalent in males, or their effects are rather more promptly manifested due to the lack of the backup X chromosome.

Authors response: We agree with the Reviewer. The sentence with that statement has been removed in the revised version of the text.

Reviewer 2: Lines 1276-1278: Looks like BACE1 is palmitoylated at 4 cysteine residues and subsequently becomes a substrate for ZDHHC3 etc. Is it true?

Authors response: We thank Reviewer for noting that. We provide information on the sites of palmitoylation of BACE1 protein and identified ZDHHCs for which BACE1 is a substrate. However the word “becomes” was misused. In the revised version of the text, we have rephrased that sentence:In addition, BACE1 is a substrate for ZDHHC3, ZDHHC4, ZDHHC7, ZDHHC15 and ZDHHC20 and may be palmitoylated at four cysteine residues: C474, C478, C482, and C485 [260]”.

In addition all the following minor comments were grouped and addressed in the same way:

Whenever a protein is mentioned for the first time, the full name need to be provided, like for PRG-1, GaS, SNAP, GRIP, TC-10 and so on.

Line 28: there are more than 300 types of PTMs in cell, not 300 PTMs.

Lines 31-33: it is hard to agree that acylation and oxidation based modification is more complex than SUMOylation or polyubiquitination…  please rearrange the sentence.

Line 69: every type of lipid modification is catalysed by (instead of “has its”) specific enzymes (unique is not precise here since enzymes actions may be redundant)

Figure 1 should be referred to when PATs mode of action is described (line 88 and further)

Lines 103-105: S-PATs are recruited to the cell membrane (and not to the cell surface), they are transmembrane proteins with function inside the cell.

Lines 199-202: “In contrast” rather than “in addition”, since NtBuHA mimics action of PPTs/ATPs so net effect of this compound on palmitoylation level is opposite to that obtained when inhibitors are used.

Line 385: in small … instead of “as small…”?

Line 956: GluN2b instead of Glu2b

Line 1040: loss-of-function mutation is correct definition

Line 1139: between young and old mice

Lines 1361-1362: ZDHHC8 simply codes for PAT, instead “which has a significant function of PAT production”

Authors response: We have corrected the text as requested by the Reviewer. Corrections are highlighted in bold in the revised version of the manuscript.